# Bedrock geology controls on new water fractions and catchment functioning in contrasted nested catchments

Guilhem Türk[1,2*], Christoph J. Gey [3], Bernd R. Schöne [3], Marius G. Floriancic [4,5], James W. Kirchner [5,6,7], Loic Leonard [1], Laurent Gourdol [1], Richard F. Keim [1], Laurent Pfister [1,2]

[1]CATchment and ecohydrology research group, Environmental sensing and modelling unit, Luxembourg Institute of Science and Technology, Belvaux, Luxembourg
[2]Faculty of Science, Technology and Medicine, University of Luxembourg, Belval, Luxembourg
[3]Institute of Geosciences, University of Mainz, Mainz, Germany
[4]Dept. of Civil, Environmental and Geomatic Engineering ETH Zürich, Zürich, Switzerland
[5]Dept. of Environmental Systems Science, 5 ETH Zürich, Zürich, Switzerland
[6]Swiss Federal Research Institute WSL, Birmensdorf, Switzerland
[7]Dept. of Earth and Planetary Science, University of California, Berkeley, CA, USA

*Correspondence to*: Guilhem Türk (guilhem.turk@hotmail.com) or Laurent Pfister (laurent.pfister@list.lu)

**Abstract.** We still lack substantial understanding on how landscape characteristics shape the storage and release of water at the catchment scale. Here we use 13 years of fortnightly precipitation and streamflow $\delta^{18}O$ measurements together with hydrometeorological data from 12 nested catchments (0.5 to 247.5 km$^2$) in the Alzette River basin (Luxembourg) to study bedrock geology and land cover controls on streamflow generation. Streamflow responses to precipitation were highly variable. Runoff coefficients ($R_c$) were typically higher in catchments dominated by less permeable bedrock (i.e., marls and claystones, $R_c$ = 0.43 to 0.52) than in catchments with a high fraction of permeable bedrock (i.e., sandstones and conglomerates, $R_c$ = 0.19 to 0.40). The fraction of new water ($F_{new}$, water younger than ∼ 16 days in this study) determined via ensemble hydrograph separation was strongly related to differences in bedrock geology. $F_{new}$ was highest in impermeable bedrock catchments (i.e., with a dominance of marls and claystone, $F_{new}$ = 4.5 to 11.9%), increasing with higher specific daily streamflow ($F_{new}$ up to 45% in one catchment). In catchments with an important fraction of permeable sandstone and conglomerates, high $F_{new}$ variability with specific streamflow ($F_{new}$ as high as 25% in one catchment) was also found, despite a damped and delayed hydrograph response to precipitation and low $F_{new}$ (means of 1.3 to 2.7%). In the weathered bedrock catchments (i.e., dominated by schists and quartzites), rapid infiltration led to large fractions of water that was older than 12 weeks (∼ 80%) and very small fractions of water younger than two weeks (∼ 3.5%). $F_{new}$ variability with streamflow was near zero, contrasting with the rapid response of the hydrograph to precipitation events. At high specific streamflow, $F_{new}$ was also correlated with bedrock geology and certain land use types. The extensive data set of streamflow $\delta^{18}O$ enabled us to link water storage and release to bedrock geology. Such information is key for a better anticipation of water storage and release functions under changing climate conditions, i.e., long dry spells and high-intensity precipitation events.

## 1 Introduction

The controls on physical processes that dominate water storage and release are not well understood, causing large uncertainties in predictions for ungauged basins or catchment functioning under a changing climate, i.e., the intensification of the hydrological cycle (Allen and Ingram, 2002). The complex interplay of water storage and release processes in the subsurface remains an important knowledge gap which prevents accurate streamflow predictions (Fan et al., 2019). A contemporary example, illustrating the importance of spatial heterogeneity of the subsurface water storage capacity, is a study in a karst watershed in China, where specifically accounting for the effective contributing area dynamically expanding with antecedent rainfall significantly enhanced the accuracy of streamflow predictions (Ren et al., 2024). Although research on catchment processes has substantially advanced our understanding of the underlying mechanisms generating large streamflow events (Beven, 2012), predicting catchment functioning in a changing climate remains subject to considerable uncertainty (Schaefli

et al., 2011; Blöschl, 2022). Similar challenges arise for predictions in ungauged basins, or basins with attributes evolving over time (Hrachowitz et al., 2013; Fan et al., 2019). We still lack the ability to generalize findings from instrumented basins to ungauged regions, or from small experimental headwater catchment studies to larger river systems (McDonnell et al., 2007; Benettin, et al., 2022). The hydrological community is increasingly tackling this issue with advanced methods, such as regionalization-strategy-guided models for flood forecasting in ungauged catchments (e.g., Ye et al., 2024). Yet, challenges partially arise from our limited understanding of controls on streamflow response. For example, an increase of floods in large rivers is not necessarily linked to extreme precipitation alone but also to antecedent wetness (Berghuijs et al., 2016). The spatial scale is crucially important when assessing the effects of climate extremes on streamflow (Blöschl, 2022). More efforts are needed to investigate how landscape features control the spatial and temporal variability in catchment functions and storage-release processes. To do so we need to move beyond the *status quo* consisting of detailed characterizations of process heterogeneity and complexity in individual research catchments and instead search for organizing principles underlying the heterogeneity and complexity (Zehe et al., 2014).

Topography and bedrock geology are important factors controlling the magnitude and age of water storage and release in catchments (Creutzfeldt et al., 2014; Sayama et al., 2011; Tromp-Van Meerveld and McDonnell, 2006). Several studies have related these catchment attributes to streamflow generation, e.g., hydrological functioning has been related to bedrock permeability, typically showing that bedrock aquifer storage and release increase with bedrock permeability (e.g., Katsuyama et al., 2005; Uchida et al., 2006; Masaoka et al., 2021; Asano et al., 2022; Weedon et al., 2023). Other tracer-based studies in multiple catchments in Switzerland, Germany, Italy and Austria (Allen et al., 2019; Von Freyberg et al., 2018; Lutz et al., 2018; Gentile et al. 2023; Floriancic et al. 2024a) have concluded that elevation, relief, drainage density, quaternary deposits and bedrock geology are important controls on streamflow generation in mountainous environments. In Luxembourg, Pfister et al. (2017) found that the fraction of impermeable bedrock is positively correlated with the standard deviation of isotopic composition of baseflow as well as with the damping of seasonal isotopic cycles between precipitation and streamflow, suggesting shorter transit times in catchments with impermeable bedrock. Similarly, Douinot et al. (2022) and Kaplan et al. (2022) have shown slow and delayed streamflow response in catchments characterized by sandstone bedrock and more flashy streamflow response in catchments dominated by marls.

Catchment storage and release of water is temporally variable. Currently, little is known about the dominant factors that drive these temporal differences (Kirchner, 2006; Benettin et al., 2022), complicating the quantification of the changing pulse of rivers (Slater and Wilby, 2017, Blöschl et al. 2017). Hence, understanding the non-stationary drivers of streamflow generation remains a key challenge in hydrology. Although computational capabilities and the complexity of numerical models are increasing rapidly (Blöschl et al., 2014), we still fail to capture catchment functioning under unusual conditions, especially extremes (e.g., long-term droughts, extreme flood events). A major reason for this is that catchment functioning is nonstationary and non-linear (e.g., Creutzfeld et al., 2014; Maneta et al., 2018) and the mechanisms generating streamflow – especially during hydrological extreme events – are diverse (Berghuijs et al., 2016). For example, the time scales of precipitation reaching the stream vary across the year and with antecedent wetness. In a recent study, Knapp et al. (2025) described how hydrological response and transport, characterizing distinct catchment behaviours, are affected differently by precipitation intensity and antecedent wetness. Along similar lines, Knapp et al. (2019) showed that transit time distributions (TTDs) differ between seasons and with antecedent wetness, while Floriancic et al. (2024b) showed that streamflow and soil waters contain slightly more recent precipitation with wetter antecedent conditions. A better understanding of the temporally variable controls of streamflow generation would help to explain temporal variabilities in streamflow magnitude and storage release.

Many field investigations of water storage, transport and mixing processes in the subsurface have relied on input-output relationships of tracer time series. Naturally occurring stable isotopes in water exhibit nearly conservative behaviour (McGuire and McDonnell, 2007), making them powerful tools to assess water ages in streamflow (i.e., transit times of precipitation water

through the catchment to reach the stream). The major aim of this study is to leverage measurements of the stable isotope composition of water across multiple catchments to establish relationships between metrics of catchment functioning (i.e., the release of water to streamflow), and landscape attributes (e.g., catchment geological properties). Understanding the physical processes behind streamflow generation is important for realistic hydrological modelling (Seibert and McDonnell, 2015; Schaefli et al., 2011) and improving this understanding depends critically on field observations.

However, isotope-based studies have often relied on convolution or sine wave fitting approaches that are not well suited to capture the spatial and temporal heterogeneities that dominate streamflow generation in most catchments (Kirchner, 2016a, b). A common source of bias is *a priori* conjectures concerning the shape of the TTD (Remondi et al., 2018), resulting in, e.g., increasing uncertainty in mean transit time (MTT) estimates when MTT exceeds several years (DeWalle et al., 1997). More recently, calculations of the fraction of young water (Kirchner, 2016b) and transit times extracted from storage selection functions (SAS) (Benettin et al., 2015; Harman, 2015; Rinaldo et al., 2015) have been proposed as more robust methods than traditional MTT estimates. Still, SAS functions also rely on unverified assumptions regarding age distributions in catchment storage or evapotranspiration (Kirchner, 2019). The newly developed ensemble hydrograph separation (EHS) technique offers a reduced-assumption alternative, based on the linear regression of tracer concentrations in the stream and lagged concentrations in precipitation across ensembles of rainfall-runoff events (Kirchner, 2019). Young water fractions (Kirchner, 2016b) quantify contributions to streamflow with transit times less than a fixed threshold of ca. 2 to 3 months, but EHS (Kirchner, 2019) is more flexible, quantifying contributions of new water with transit times as short as the interval between stream tracer samples, or any multiple of that sampling interval. Owing to its mathematical formulation, EHS can also be applied to subsets of the tracer time series, e.g., to assess catchment functions under specific conditions or hydrologic extremes (e.g., low flow or high flow events). Following a proof-of-concept study (Knapp et al., 2019), EHS has been used to quantify mobilization of subsurface water in burned and unburned hillslopes in California (Atwood et al., 2023), to estimate new water contributions to tree xylem water in a Peruvian cloud forest (Burt et al., 2023), to identify contrasting catchment release patterns related to hydroclimatic and physical catchment properties across the Austrian and Swiss Alps (Floriancic et al., 2024a) and to assess the fractions of recent precipitation in streamflow and at different soil depths along a forested hillslope (Floriancic et al., 2024b).

This study investigates the influence of bedrock geology on the time scale of water transport in 12 nested catchments of the Alzette River basin (Luxembourg), a basin where numerous studies on hydrological functioning have been carried out in the past decade (Wrede et al., 2015; Pfister et al., 2017; Douinot et al., 2022; Kaplan et al., 2022). Located at the northern edge of the Paris Sedimentary basin, the geologically diverse Alzette River basin is an ideal setting to test the hypothesis that bedrock geology controls hydrological functioning and the time scales over which precipitation is transformed into streamflow. Our study builds on extensive long-term precipitation, hydrometric and isotope time series, including up to 13 years of fortnightly $\delta^{18}O$ records in 12 nested catchments. We use EHS to *i)* assess how precipitation intensity affects streamflow and transit times across multiple nested catchments, and *ii)* explore whether fast transit times are associated with specific types of bedrock geology.

## 2 Methods

### 2.1 Physiographic characterisation of the nested catchments

The Alzette River basin, located at the northern edge of the Paris Sedimentary Basin, spans a wide range of physiographic settings, making it suitable for studying differences in catchment functioning in contrasting nested headwater catchments. Luxembourg is dominated by schists in the north (*Oesling*) and sedimentary bedrock (marls, calcareous rocks, and sandstone) in the south (*Gutland*) (Fig.1). The northern tip of the study basin belongs to the schistose Ardennes Massif, an elevated plateau (averaging 450 m a.s.l.), where streams are incised, forming relatively steep slopes covered by forests (dominated by oak,

beech, and spruce), whereas the plateaus are mainly covered by grassland and cropland. In the remaining part of the basin, sandstone and limestone lithologies are mainly covered by forests, while the marl and claystone catchments are mainly covered by cropland or grassland. Three of the 12 nested catchments have been intensively monitored since 2000 (Wrede et al., 2015): the Weierbach (0.45 km$^2$), Wollefsbach (4.5 km$^2$), and Huewelerbach (2.7 km$^2$). These three catchments have been studied in detail in previous studies (Wrede et al., 2015; Martínez-Carreras et al., 2016; Pfister et al., 2017; Kaplan et al., 2022) and we rely on these in-depth catchment descriptions to infer the behaviour and characteristics of the other nine catchments in similar geologic settings:

The *Weierbach* is a steep, forested catchment in the schistose Ardennes massif with elevation rangeing from 422 to 512 m a.s.l. It has been monitored and sampled for more than 20 years mainly for eco-hydrological studies (Hissler et al., 2021). The bedrock has generally low porosity, with higher porosities in cracks and fissures. The Weierbach catchment is covered by loamy soils formed from weathered regolith and periglacial deposits, filling the bedrock cracks and fissures and creating substantial subsurface storage volumes. High degrees of weathering and deep root structures enhance infiltration rates and drainage along preferential flow paths, which explains the dominance of subsurface flow in saprolite and periglacial deposits (Angermann et al., 2017). Another characteristic of the catchment is the seasonal difference in streamflow response to precipitation. In summer, streamflow reacts concomitant with rainfall, while in winter there are typically two peaks, one immediately in response to a precipitation event and a second delayed streamflow peak occurring several hours to days after the event (Martínez-Carreras et al., 2016; Pfister et al., 2023).

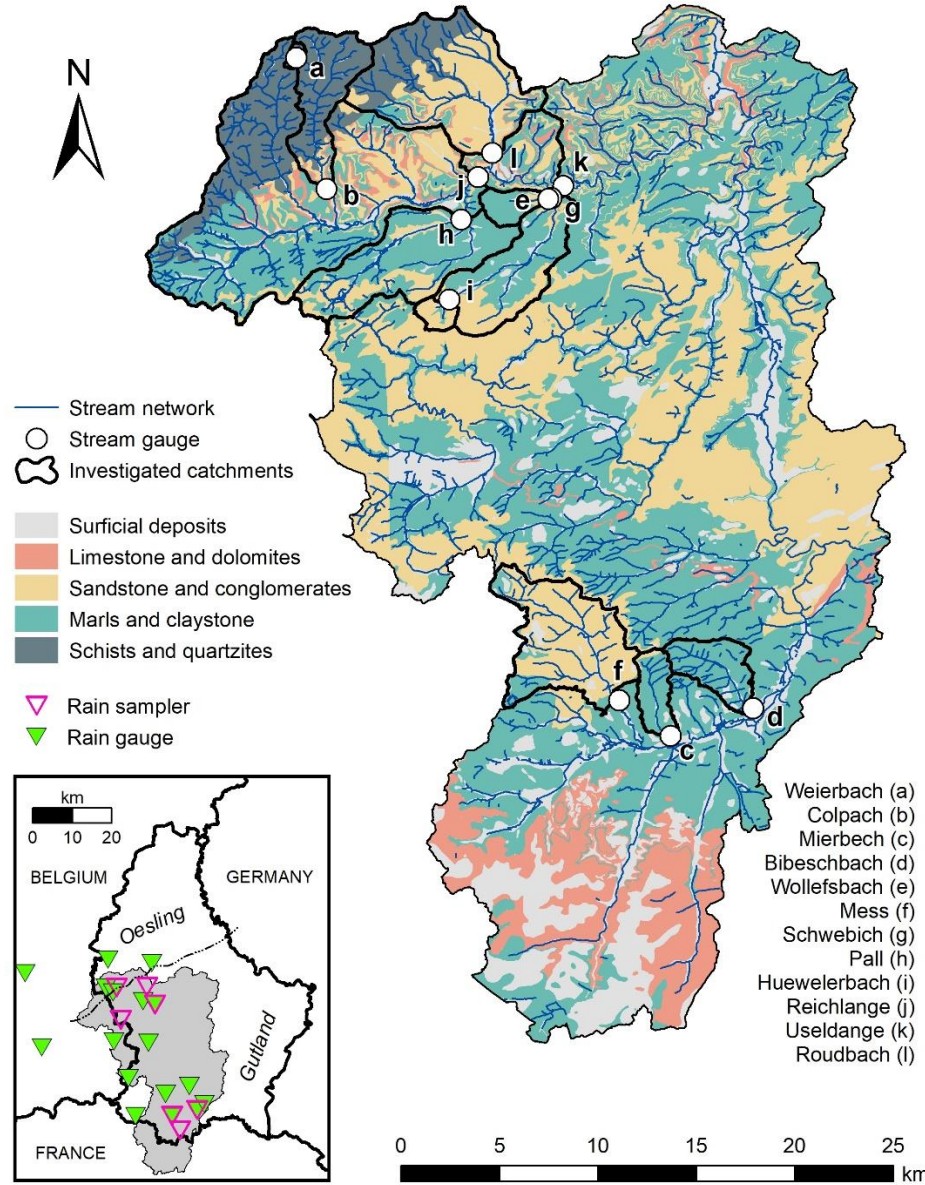

**Figure 1: Large map: geologic context of the Alzette River basin with the stream network, the boundaries of the nested catchments, and the location of the stream gauges. Box: extent of the Alzette River basin (grey shaded area) relative to the boundaries of Luxembourg, Belgium and France (grey lines), and locations of the rain samplers (for isotopic measurements) and rain gauges. Stream samples were taken next to the stream gauges. Geologic maps of Luxembourg and the study area were provided by the Luxembourgish geologic survey (APC, 2022), and other sources for areas of the Alzette River basin outside of the Luxembourgish boundaries (Hellebrand et al., 2007; BRGM, 2020). The lithology was simplified by regrouping geologies with similar properties for the hydrological characterisation of the catchments (Nijzink et al., 2024).**

The *Wollefsbach* catchment is formed by alternating layers of marl, sandstone, and limestone inclined southwards (Wrede et al., 2015). The upstream section of the Wollefsbach is dominated by mostly impermeable sandy marls. The gently undulating landscape with elevations ranging from 245 to 306 m a.s.l. is dominated by croplands and pastures, with a dense drainage network that was artificially expanded to avoid cropland waterlogging. The loamy soils, with generally low permeability, have macropores from cracks that become established during extended summer dry spells. Macropores of biogenic origin are widespread in spring and autumn because of high earthworm abundance, but are partly disconnected by ploughing, shrinking and swelling of the soils. The seasonal changes in dominant runoff generation processes (i.e., overland flow, shallow

subsurface flow in cracks and macropores), in combination with the dense artificial subsurface drainage networks, ultimately lead to a flashy runoff regime (Loritz et al., 2017).

The *Huewelerbach* catchment is typical for the Luxembourg sandstone cuesta landscape, lying on top of a sloped marly bedrock formation (Pfister et al., 2018). Thus, alternating layers of porous sandstone and calcareous sandstone are found on the plateaus and on top of the hillslopes, while alternations of marls and limestone emerge in the lower parts of the hillslopes

and in sections of the valleys close to the stream. The alternation of highly permeable sandstone and, in comparison, less permeable calcareous sandstone layers results in a heterogeneous, yet high, hydraulic conductivity (on mean, $5 \times 10^{-5}$ m s$^{-1}$; Gourdol et al., 2024). The catchment is dominated by a steep central valley with elevations ranging from 280 to 400 m a.s.l., is forested on the plateaus and hillslopes and is covered by grasslands on the foot slopes. The sandy, highly permeable soils prevent surface runoff in large parts of the catchment, except for some downstream sections of the valley that are underlain by

marls. The permeable sandstone lying on top of the much less permeable layer of marls provides large aquifer storage with a seasonally constant groundwater flow feeding the perennial springs located at the interface between the two layers (Fenicia et al., 2014).

The other nine nested catchments in our study area range in size from 4.4 km$^2$ to 247.5 km$^2$ and have similar geologies compared to the Weierbach, Huewelerbach, and Wollefsbach catchments. We grouped the catchments into four hydro-

lithological categories, depending on bedrock geology, the related permeabilities, hydraulic conductivity and storage capacity (see Table 1):

i. *Weathered layer catchments* (type Weierbach): Shallow weathered regolith on top of fractured bedrock (e.g., schists and quartzites) in steep terrain, with substantial storage capacity (i.e., Colpach catchment, 19.2 km$^2$). The river network is dense and concentrated near the valley bottom (Fig. 1), displaying both the high infiltration capacity of

the shallow weathered layer on the plateaus and hillslopes, and the impermeability of the bedrock.

ii. *Impermeable layer catchments* (type Wollefsbach): Impermeable bedrock (e.g., marls and claystone) with clayey soils, locally sandy facies, limited storage capacities and preferential flow paths, which are reflected in the dense river network (Fig. 1). This category includes the Mierbech (6.8 km$^2$) and the Bibeschbach (10.6 km$^2$) catchments. Despite being a sandstone catchment (Fig.1), the Mess (32.2 km$^2$) is also attributed to this category. The marly sandstone

(with sandy claystone) of the Mess has favoured the development of a dense river network that is more characteristic of the impermeable bedrock geologies than permeable ones (e.g., sandstone and conglomerates).

iii. *Permeable layer interface catchments* (type Huewelerbach): permeable bedrock (e.g., sandstone and conglomerates) of decreasing thickness towards the valleys with emergence of the underlying less permeable layer (e.g., marls and claystone). These catchments have large storage capacities, with springs and a less dense river network with

streamflow generation at the interface between the permeable and impermeable layer. This category includes the Pall (33 km$^2$) and Schwebich (30.1 km$^2$) catchments.

iv. *Aggregated catchments*: Upstream schistose and quartzite geologies are gradually replaced by sandstone and conglomerates or marls and claystone downstream leading to differences between upstream and downstream streamflow generation processes. This category includes the Roudbach (44 km$^2$) catchment and the larger Attert River

basin subcatchments with gauges at Reichlange (159.5 km$^2$) and Useldange (247.5 km$^2$).

**Table 1: Catchment area, mean elevation, fraction of major land use types (forest, grassland, agriculture), fraction of bedrock geology types (limestone, sandstone or conglomerates, marls or claystone and schists or quartzites) and fraction of alluvial deposit cover for each of the 12 sub-catchments of the Alzette River basin, extracted from the CAMELS-LUX database (Nijzink et al., 2024).**
**The hydro-lithological categories given in the last column are as follows: weathered layer (WL), impermeable layer (IL), permeable layer interface (PLI), aggregated (AG) catchments.**

| Station | Area [km²] | Elevation [m asl.] | Forest [%] | Grassland [%] | Agriculture [%] | Limestone [%] | Sandstone [%] | Marls [%] | Schists [%] | Alluvial [%] | Categories |
|---|---|---|---|---|---|---|---|---|---|---|---|
| a Weierbach | 0.5 | 497 | 95.6 | 0.0 | 0.0 | 0.0 | 0.0 | 0.0 | 97.4 | 2.6 | WL |
| b Colpach | 19.2 | 442 | 51.0 | 24.0 | 23.0 | 5.0 | 1.0 | 10.0 | 81.0 | 3.0 | WL |
| c Mierbech | 6.8 | 310 | 34.9 | 29.1 | 30.0 | 0.0 | 6.1 | 85.1 | 0.0 | 8.7 | IL |
| d Bibeschbach | 10.6 | 298 | 49.5 | 22.4 | 18.5 | 0.0 | 0.3 | 90.1 | 0.0 | 9.5 | IL |
| e Wollefsbach | 4.4 | 279 | 7.1 | 51.3 | 38.7 | 0.6 | 10.7 | 81.8 | 0.0 | 6.9 | IL |
| f Mess | 32.2 | 325 | 12.2 | 41.5 | 33.2 | 0.0 | 75.7 | 12.7 | 0.0 | 11.6 | IL |
| g Schwebich | 30.1 | 297 | 30.6 | 40.1 | 23.6 | 0.3 | 33.5 | 56.3 | 0.0 | 9.9 | PLI |
| h Pall | 33.0 | 310 | 22.7 | 43.6 | 26.3 | 0.0 | 24.6 | 64.3 | 0.0 | 11.1 | PLI |
| i Huewelerbach | 2.7 | 353 | 89.6 | 5.7 | 0.1 | 0.0 | 81.5 | 11.8 | 0.0 | 6.7 | PLI |
| j Reichlange | 159.5 | 357 | 31.8 | 30.5 | 30.8 | 5.3 | 18.1 | 40.6 | 27.8 | 8.2 | AG |
| k Useldange | 247.5 | 353 | 31.5 | 30.1 | 31.2 | 4.9 | 26.2 | 36.8 | 23.9 | 8.2 | AG |
| l Roudbach | 44.0 | 396 | 36.0 | 20.5 | 35.3 | 3.4 | 49.3 | 8.2 | 33.3 | 5.7 | AG |

## 2.2 Hydrometric variables in the nested catchments

Our experimental set-up consisted of 12 river gauges and 16 rain gauges (Fig. 1). Hourly precipitation measurements from the LIST monitoring platform were obtained from June 2011 to March 2023 and spatially interpolated separately between the northern (n = 10) and southern stations (n = 7), using ordinary Kriging (*fit.variogram* and *krige* functions, *gstat* package in R; Pebesma, 2004; Gräler et al., 2016). Hourly specific streamflow measurements were also obtained from the LIST monitoring platform from June 2011 to March 2023. We took hourly specific streamflow to calculate daily mean values, taking the 24 hours preceding grab samplings in the streams. Some time series were not continuous due to technical issues. The summer 2021 flood event destroyed the Reichlange station, while the relationship between stage and streamflow changed at Mess and Pall due to changes in river morphology. These missing streamflow values (a total of 622 missing values across all 12 stations out of 56.152 values in total) were interpolated as follows: at each station, we calculated the long-term mean for each day of the year and the ratio between the daily long-term mean and the actual value. To estimate the missing streamflow at a station, we used the ratio from neighbouring stations and multiplied it with the daily long-term mean. Monthly potential evapotranspiration (PET) was obtained using the Thornthwaite (1948) approach from monthly air temperature and day length, as per Pfister et al. (2017).

Our dataset covers 13 years of hydrological observation, encompassing extremely wet (e.g., floods in January 2011, May 2013 and July 2021) and dry intervals (e.g., spring drought 2011; Fig. 2) and some of the warmest years on record (2011, 2017, 2018, 2019, 2020 and 2021 each broke monthly temperature records, MeteoLux (https://www.meteolux.lu/fr/climat/normales-et-extremes/), accessed on 14/03/2025). The study area is characterised by a precipitation gradient from northwest to southeast, typical for the region (Pfister et al., 2004), with annual precipitation ($P_a$) ranging from 714 mm in the Bibeschbach catchment to 913 mm near Reichlange (Table 2). Precipitation was rather evenly distributed across the seasons, typically with highest totals in winter (265 mm ± 74 mm), similar totals in summer and autumn (206 mm ± 65 mm and 198 mm ± 71 mm, respectively) and lowest totals in spring (154 mm ± 54 mm). Annual potential evapotranspiration ($PET_a$) varied little between catchments,

ranging from 607 to 667 mm and showed pronounced seasonality. Potential evapotranspiration totals were typically highest in summer (272 mm ± 70 mm), similar in spring and autumn (148 mm ± 39 mm and 123 mm ± 32 mm, respectively) and lowest in winter (61 mm ± 16 mm).

## 2.3 Calculation of storage dynamics

We used precipitation, streamflow and potential evapotranspiration values to calculate catchment storage $S(t)$ in mm as per Pfister et al. (2017):

$$S(t) = [R(t) - Q(t) - \alpha E(t)] + S(t-1), \qquad (1)$$

$$if\ S(t-1) < 200\ mm, \alpha = S(t-1)/200$$

$$if\ S(t-1) > 200\ mm, \alpha = 1$$

where $S(t)$ is catchment storage (in mm) at day $t$, $R(t)$ daily precipitation (mm d$^{-1}$), $Q(t)$ is daily streamflow (mm d$^{-1}$), $E(t)$ is daily potential evapotranspiration (mm d$^{-1}$), and $\alpha$ is a weighting coefficient that limits evapotranspiration with decreasing water availability. Daily PET was obtained by dividing the monthly PET by the numbers of days inside a month, described as a suitable approximation in Pfister et al. (2017). We estimated the maximum storage capacity ($S_{max}$) for each catchment, defined as the highest 0.5 % of daily catchment storage values and computed the storage deficit $D(t)$, as follows:

$$D(t) = S_{max} - S(t). \qquad (2)$$

For the weighing factor $\alpha$ we assume that field capacity is reached at a catchment storage threshold of 200 mm, as suggested in Pfister et al. (2017). They assessed the sensitivity of the storage deficit estimates for different values of the field capacity (100, 200, 300 mm) and found the daily offsets to be largely unaffected by the value chosen for field capacity. Consequently, although the absolute storage estimates might differ, the storage deficit, used in this study, remains unaffected. In some cases,

the exact water balance was difficult to estimate, because catchments delineated based on topographic criteria might not be entirely representative of the actual drainage area, notably when the topographic boundaries of the aquiclude layer do not exactly follow surface topography. Closing the water balance in the Huewelerbach additionally required accounting for water withdrawn for drinking water from springs in the catchment, on top of likely subsurface flow out of the catchment due to tilting of the geological layers. To approximate these, we assumed a constant loss of 0.4 mm d$^{-1}$ to match the lower end of runoff

ratios in the other catchments ($R_c$ = 0.35).

## 2.4 Stable water isotope sampling

Cumulative precipitation was sampled every 2 weeks for analysis of the stable isotope composition at four rain gauges in the northern part and three rain gauges in the southern part of the study area (Fig. 1). Similarly fortnightly grab samples of

streamflow were taken at the outlet of each of the 12 nested catchments between June 2011 and March 2023. All precipitation and streamflow samples were analysed for $\delta^{18}O$ and $\delta^2H$ using an off-axis integrated cavity output laser spectrometer (Los Gatos TIWA-45-EP, OA-ICOS). Values are reported in per mil notation relative to the Vienna Standard Mean Ocean Water 2 standard (VSMOW2) (IAEA, 2017) with a nominal accuracy of 0.2 ‰ for $\delta^{18}O$ and 0.5 ‰ for $\delta^2H$. Due to gaps in the time series or delayed sampling, the sampling interval was not strictly 14 days, but ~16 days across the entire time series. If intervals

between two samplings were longer than 1 month (30 days), they were excluded from the EHS analyses. Samples with precipitation below 0.5 mm were also removed from the analyses.

**2.5 Ensemble hydrograph separation and fractions of new water**

We used the hydrometric and isotope time series to calculate the fraction of new water ($F_{new}$) in each sample with the ensemble hydrograph separation (EHS) method introduced by Kirchner (2019). EHS estimates the mean contribution of precipitation that becomes streamflow within a time scale defined by the sampling interval. In classical hydrograph separation, $F_{new,j}$ is a simple expression of differences in tracer concentrations in streamflow before the event, precipitation (or event water), and streamflow during the event, based on a simple mass balance (Pinder and Jones, 1969; Rodhe, 1987):

$$F_{new,j} = \frac{c_{Q,j} - c_{Q,j-1}}{c_{P,j} - c_{Q,j-1}}, \tag{3}$$

where $C_{Q,j}$, $C_{Q,j-1}$ and $C_{P,j}$ are the tracer concentrations in the stream after, before, and the event water (i.e., precipitation). The underlying idea of EHS is that Equation 3 can also be expressed as a linear regression, where $F_{new}$ is the regression slope in a scatterplot of $C_{Q,j}$ - $C_{Q,j-1}$ versus $C_{P,j}$ - $C_{Q,j-1}$ over the ensemble of observations $j$ (Kirchner, 2019):

$$C_{Q,j} - C_{Q,j-1} = F_{new}(C_{P,j} - C_{Q,j-1}) + \alpha + \varepsilon_j, \tag{4}$$

where $\alpha$ is the regression intercept and $\varepsilon_j$ is the error term. The major difference here is that $F_{new}$ is now a single value, representing an ensemble average of the time-variable $F_{new,j}$, weighted by the isotopic difference between event and pre-event water, and thus by the reliability of estimating each $F_{new,j}$ in Equation (1). The regression slope $F_{new}$ can be calculated as:

$$F_{new} = \frac{cov(C_{P,j}-C_{Q,j-1},\ C_{Q,j}-C_{Q,j-1})}{var(C_{Q,j}-C_{Q,j-1})}. \tag{5}$$

Next, contributions of precipitation to streamflow can be estimated over a range of lag times $k$, with a maximum lag time $m$, to resolve the transit time distribution (TTD) in the catchment. Eventually one is left with an expression analogous to Equation 4:

$$C_{Q,j} - C_{Q,j-m-1} = \sum_{k=0}^{m} \beta_k(C_{P,j-k} - C_{Q,j-m-1}) + \alpha + \varepsilon_j, \tag{6}$$

where $\beta_k$ is the fraction of streamflow exiting at time $j$ and that had entered $k$ time steps earlier. The regression estimates of $\beta_k$ can then be used to estimate the TTD, with the solution provided in Kirchner (2019).

Note that the calculation of $F_{new}$ *per se* excludes sampling intervals in which no rain fell, and that they need to be separately accounted for using a correction factor (not applied in this study, because few of our ~16-day sampling intervals were rainless, merely 16 days across all catchments). EHS also assumes that the sampling interval is constant, as it compares isotopic signatures at a certain timestep with previous observations without considering the exact time of sampling. Nonetheless EHS can be applied to time series with somewhat unequal timesteps. In this case, the best approximation is to take the mean sampling interval as the definition for the fraction of water that is "new", as we did in this study (i.e., 16 days). Gaps in isotopic measurements need to be filled with NA values, while zeros can be used to fill gaps in streamflow (if $F_{new}$ calculations are flow-weighted) or precipitation records. This will not affect the computations if a precipitation threshold higher than zero is set for the computations (here 0.5 mm). It is important to fill in missing values as described above, rather than simply removing those records from the time series, because EHS assumes a link between two consecutive samples in the record, which could otherwise be multiple sampling intervals apart. The sampling interval in this study is approximately ~16 days, thus EHS estimates based on the scripts in R (or Matlab) provided by Kirchner and Knapp (2020) allowed us to calculate how much of streamflow is younger than ~16 days. The $F_{new}$ results presented here are based on the robust estimation algorithm (i.e., less sensitive to outliers), unless designated otherwise, and are not weighted by streamflow amount. A major advantage of using EHS is that it can be applied to subsets of the tracer time series, which we leverage to create profiles (see Kirchner and Knapp, 2020) of the fraction of water with fast flow paths as functions of increasing precipitation and streamflow rates.

The calculated interval of ~16 days for the definition of $F_{new}$ does not allow us to assess certain fast-response components that would require higher temporal resolutions. During high streamflow events, much of the streamflow response will occur within that two-week time window. For example, once either the infiltration capacity or the storage capacity of the soils in a catchment is exceeded, overland flow processes may occur very quickly, within hours or even minutes (Blöschl, 2022; Douinot et al.,

2022). Still, because of the integrative nature of $F_{new}$, these fast-response components are also represented by $F_{new}$ in this study, as they represent part of the water with age 0 to 16 days. In this context, our study could serve as a guide to identify streams that would benefit from high-frequency sampling or monitoring campaigns. It is also important to consider that when the sampling campaign in the Alzette River basin started in 2011, EHS had not yet been developed. Often, monthly data would suffice to calculate metrics such as, e.g., water travelling to the steams in less than 2-3 months, based on convolution or sine wave fitting techniques (Kirchner, 2016). This had been done in analogous inter-catchment studies in Germany (Lutz et al., 2018) and in Switzerland (Von Freyberg et al., 2018), relying on monthly, or fortnightly isotope data. In this regard, fortnightly isotopic measurements in multiple nested catchments in the Alzette River basin represented a state-of-the-art dataset at the time the sampling began. We argue that moving from the previous definition of fractions of water less than 2-3 months (i.e., young water fractions) to fractions of water less than ~16 days old (i.e., ensemble hydrograph separation) represents a considerable step forward in characterizing water storage and transport in these catchments.

## 3 Results

### 3.1 Storage-release dynamics

Although annual precipitation totals were similar across the entire Alzette River basin and roughly distributed evenly across seasons, streamflow was much more variable across the 12 nested catchments, as reflected in the hydrographs and flow duration curves (FDCs, Fig. 2). The specific streamflow shows clear seasonality with alternating dry summers and wet winters, following the seasonal variation of evapotranspiration. The seasonality in specific streamflow was less pronounced in the Roudbach and particularly weak in the Huewelerbach catchment (Fig. 2). While zero-flow conditions frequently occurred in the impermeable layer or weathered layer catchments, specific streamflow in the other catchments was rarely below ~ 0.1 mm $d^{-1}$. The interannual variability also appeared to be more pronounced in catchments that run occasionally dry than in catchments with perennial flow. Overall, catchments with similar geologies had similar FDCs.

Mean annual specific streamflow ($q_a$) was highest in the weathered layer catchments (Weierbach: 459 mm, Colpach: 453 mm), followed by the Attert at Useldange (426 mm) and Attert at Reichlange (415 mm) (Table 2). Mean annual specific streamflow in the Mess ($q_a$ = 346 mm), dominated by marly sandstones, was much higher than in the other impermeable layer catchments (mean $q_a$ ~ 250 mm). Mean annual specific streamflow of the Huewelerbach was the lowest, 161 mm. Annual streamflow totals in all nested catchments were dominated by high winter streamflow, with ratios of summer to winter streamflow $Q_{S/W}$ ranging from 0.21 to 0.38 (Table 2). An exception to this was the Huewelerbach catchment, where summer streamflow was only ~ 30% lower than winter streamflow ($Q_{S/W}$ = 0.70).

**Table 2: Hydrometric characteristics for the 12 nested catchments in the Alzette River basin: sampling start and end, annual specific streamflow, precipitation and potential evapotranspiration ($q_a$, $P_a$ and $E_a$, respectively), maximum storage capacity ($S_{max}$), runoff coefficient ($R_c$), highest 0.5 percentile of hourly specific streamflow ($q_{99.5}$), and ratio of summer versus winter streamflow ($Q_{S/W}$).**

| Station | Start | End | $q_a$ [mm] | $P_a$ [mm] | $E_a$ [mm] | $S_{max}$ [mm] | $R_c$ [-] | $q_{99.5}$ [mm h$^{-1}$] | $Q_{S/W}$ [-] |
|---|---|---|---|---|---|---|---|---|---|
| a Weierbach | 01/01/2010 | 30/08/2023 | 459 | 865 | 607 | 197 | 0.52 | 0.55 | 0.25 |
| b Colpach | 01/01/2010 | 05/01/2023 | 453 | 885 | 642 | 173 | 0.51 | 0.48 | 0.23 |
| c Mierbech | 01/01/2010 | 17/01/2023 | 243 | 736 | 657 | 192 | 0.34 | 0.50 | 0.21 |
| d Bibeschbach | 01/01/2010 | 03/01/2023 | 254 | 714 | 657 | 180 | 0.36 | 0.50 | 0.32 |
| e Wollefsbach | 01/01/2010 | 09/01/2023 | 257 | 748 | 642 | 145 | 0.34 | 0.61 | 0.22 |
| f Mess | 01/01/2010 | 14/02/2022 | 346 | 745 | 667 | 154 | 0.43 | 0.65 | 0.26 |

| | | | | | | | | |
|---|---|---|---|---|---|---|---|---|
| *g Schwebich* | 01/01/2010 | 09/01/2023 | 282 | 786 | 642 | 191 | 0.36 | 0.47 | 0.34 |
| *h Pall* | 01/01/2010 | 31/01/2022 | 378 | 905 | 642 | 243 | 0.40 | 0.54 | 0.34 |
| *i Huewelerbach* | 01/01/2010 | 04/01/2023 | 161 | 882 | 642 | 433 | 0.19 | 0.09 | 0.70 |
| *j Reichlange* | 01/01/2010 | 01/01/2021 | 415 | 913 | 641 | 197 | 0.45 | 0.47 | 0.30 |
| *k Useldange* | 01/01/2010 | 09/01/2023 | 426 | 867 | 641 | 171 | 0.49 | 0.42 | 0.33 |
| *l Roudbach* | 01/01/2010 | 09/01/2023 | 336 | 847 | 625 | 237 | 0.40 | 0.23 | 0.38 |

The largest peak hourly specific streamflow on record was in the Mess catchment ($q_{99.5}$ = 0.65 mm h$^{-1}$), followed by the Wollefsbach ($q_{99.5}$ = 0.61 mm h$^{-1}$, Table 2). In the weathered layer and impermeable layer catchments, peak streamflow ranged from 0.48 mm h$^{-1}$ to 0.65 mm h$^{-1}$. Peak streamflow was slightly lower at the Attert gauges (Reichlange and Useldange), and much lower in the Roudbach ($q_{99.5}$ = 0.23 mm h$^{-1}$), all combined in the aggregated catchment category. Comparing the permeable layer interface catchments, peak streamflow from the Pall was 0.54 mm h$^{-1}$, comparable to the Schwebich ($q_{99.5}$ = 0.47 mm h$^{-1}$), in contrast to the Huewelerbach, where peak streamflow was much lower ($q_{99.5}$ = 0.09 mm h$^{-1}$). Peak streamflow in all catchments was well synchronised (i.e., the large streamflow events occurred after the same rain events in all catchments), but magnitudes differed (Fig 2) during, e.g., the major flood in July 2021. Mean runoff coefficients ($R_c$) were typically larger in catchments dominated by less permeable bedrock (Fig. A1 in Appendix). Runoff coefficients were highest in the schistose Weierbach and Colpach catchments and the Attert in Reichlange, Attert in Useldange and the Mess catchments ($R_c$ = 0.45 to 0.52, Table 2). The runoff coefficient in the sandstone-dominated Huewelerbach was by far the lowest ($R_c$ = 0.19).

Maximum storage capacity ($S_{max}$) in the nested catchments ranged from 145 mm to 433 mm (Table 2). The highest $S_{max}$ was by far in the Huewelerbach, where the storage deficit ($D$) also varied the most (Fig. 3 and A2). The next highest $S_{max}$ occurred in the Pall ($S_{max}$ = 243 mm) and the Roudbach ($S_{max}$ = 237 mm), both containing a large percentage of sandstone facies, like the Huewelerbach. The seasonal variance in $D$ of those catchments was also high. In the remaining catchments, $S_{max}$ was similar, typically around 200 mm. We also observed seasonal variations of $D$ (Fig. A2), with storage close to the calculated maximum during winter, when streamflow was typically higher. Although there was a clear negative relationship between streamflow and storage deficit, there were also examples of high streamflow during times with large storage deficits, especially in the impermeable layer catchments, e.g., in the summer of 2013 in the Mierbech, Bibeschbach and Mess catchments.

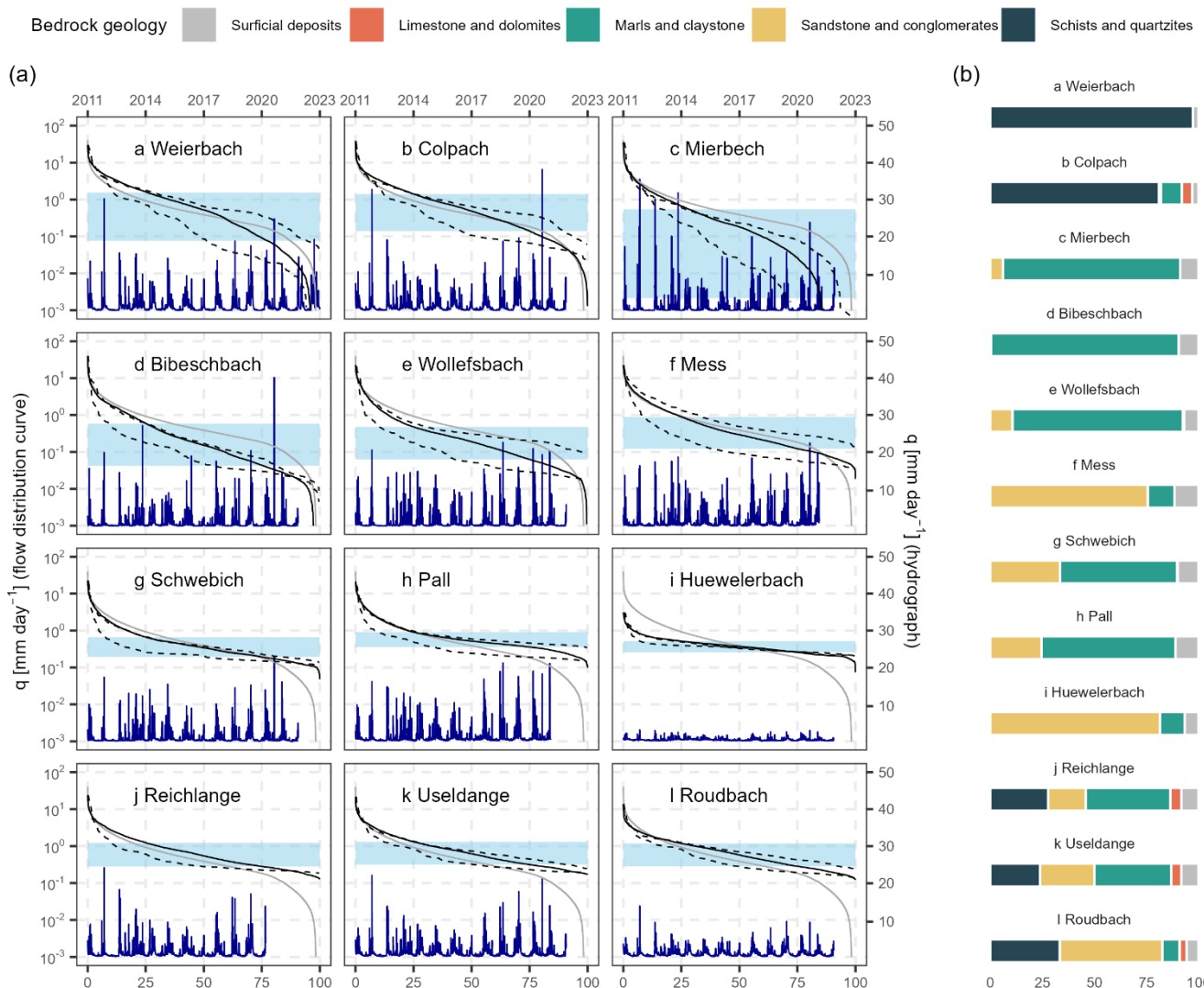

**Figure 2: (a)** Flow duration curves (FDCs – left y axis (log scale), bottom x axis) and hydrographs (right y axis, top x axis) for the 12 nested catchments. Grey lines represent the mean FDC for the entire Alzette River basin, while solid black lines represent the mean FDCs calculated using the full specific streamflow time series in the individual catchments. FDCs of some individual years are shown to highlight exceptional years (2011, lower dashed line, and 2021, upper dashed line; missing for the Reichlange catchment). The interquartile range (IQR) of the daily specific streamflow in the individual catchments is indicated in light blue shading. **(b)** Fraction of bedrock geology types in the individual catchments. The results show highly contrasting streamflow behaviours between catchments of different bedrock geologies, with overall smaller streamflow variation in catchments consisting of a larger fraction of sandstone and conglomerates.

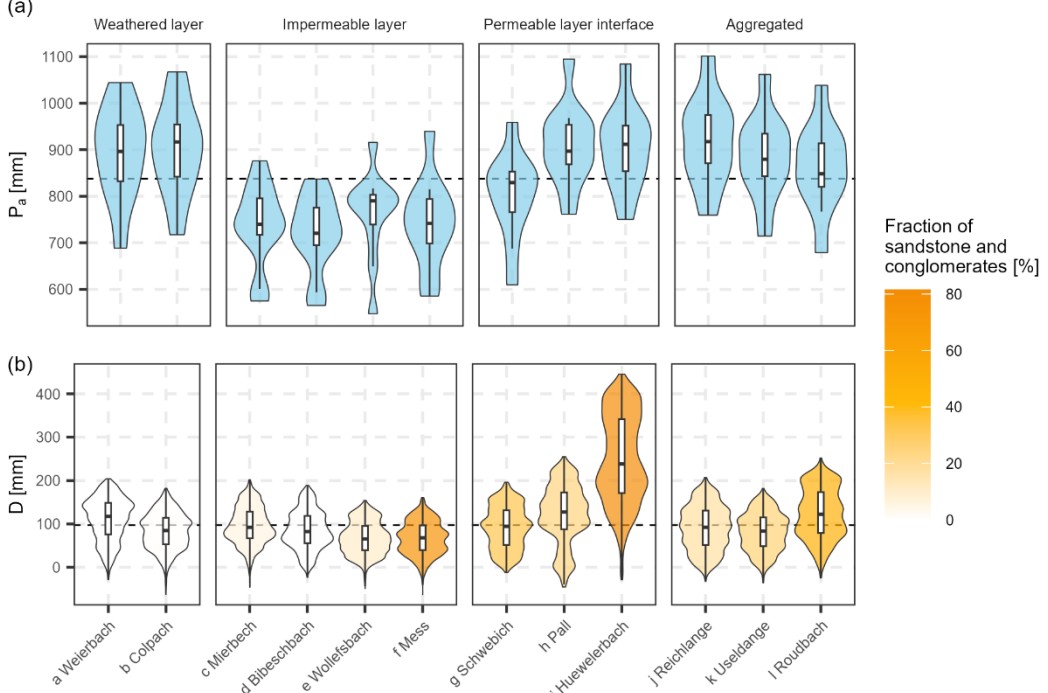

**Figure 3: Violin plots and boxplots of (a) annual precipitation ($P_a$) and (b) daily storage deficit ($D$) in the 12 nested catchments. The dashed horizontal lines represent the medians across all catchments for reference. Annual precipitation is similar across the catchments, but small differences exist according to the location of the catchments along the northwest to southeast gradient of precipitation totals. The dynamics of storage deficits were of similar range in most catchments, yet larger in catchments with a large fraction of sandstone and conglomerates.**

### 3.2 Isotope variability in fortnightly stream and precipitation measurements

Fortnightly $\delta^{18}$O of the four stations in the north and the three stations in the south followed similar patterns. Winter $\delta^{18}$O was lower than in the summer, following the typical seasonality induced by temperature expected at this latitude (Dansgaard, 1964; Feng et al., 2009) (Fig. 4). Median $\delta^{18}$O varied little between neighbouring stations, with values of -7.3 ‰ and -7.4 ‰ in the north and south, respectively, and all with interquartile ranges around 4.0 ‰.

The amplitude of the seasonal precipitation $\delta^{18}$O signal was damped in streamflow, with large differences between the different hydro-lithological catchment categories (Fig. 5). Catchments with permeable layer interfaces and large storage volumes (e.g., Huewelerbach) had much smaller seasonal streamflow $\delta^{18}$O amplitudes compared to the catchments dominated by marly bedrock (and thus less permeable subsurface properties) with limited storage capacity (Fig. 5). Streamflow $\delta^{18}$O amplitudes were also smaller in weathered layer catchments. Interquartile ranges of $\delta^{18}$O varied between 0.6 ‰, 0.8 ‰, 0.4 ‰ and 0.4 ‰ for weathered layer, impermeable layer, permeable layer interface and aggregated catchments, respectively. Streamflow $\delta^{18}$O from catchments with similar bedrock geologies showed similar seasonal amplitudes. Compared with the median $\delta^{18}$O in precipitation, the median streamflow $\delta^{18}$O was lower in the weathered layer (-8.4 ‰), aggregated (-7.9 ‰) and permeable layer interface catchments (-7.9 ‰). However, in the impermeable layer catchments, median streamflow $\delta^{18}$O (-7.4 ‰) was closer to the median $\delta^{18}$O in precipitation. Streamflow $\delta^{18}$O in catchments from all hydro-lithological categories were occasionally affected by strong rainfall, with pronounced deviations from the typical streamflow $\delta^{18}$O (Fig. 5a).

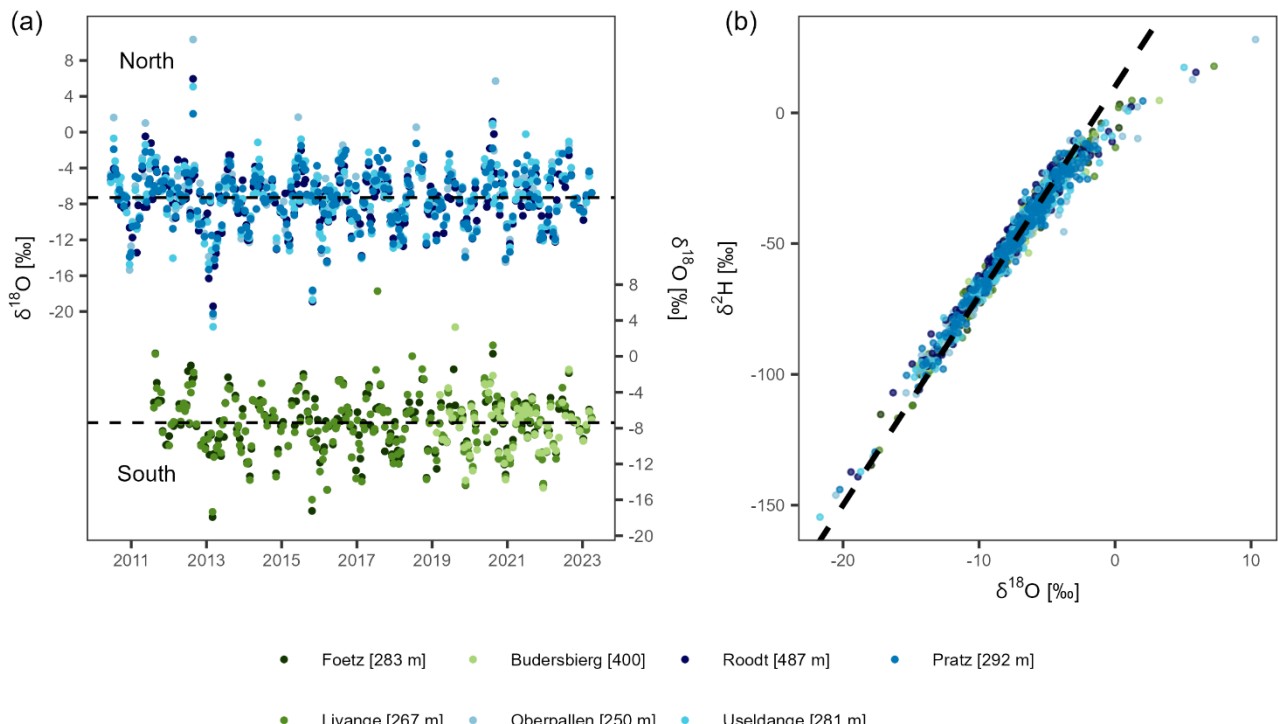

**Figure 4: (a) Oxygen stable isotope composition of fortnightly precipitation samples of the seven rain gauges between 2011 and 2023. (b) Dual isotope plot (oxygen and hydrogen) of the isotope composition at the seven rain gauges. The dashed line indicates the Global Meteoric Water Line (GMWL) with the equation $\delta^2H = 10 + 8 \times \delta^{18}O$.**

395

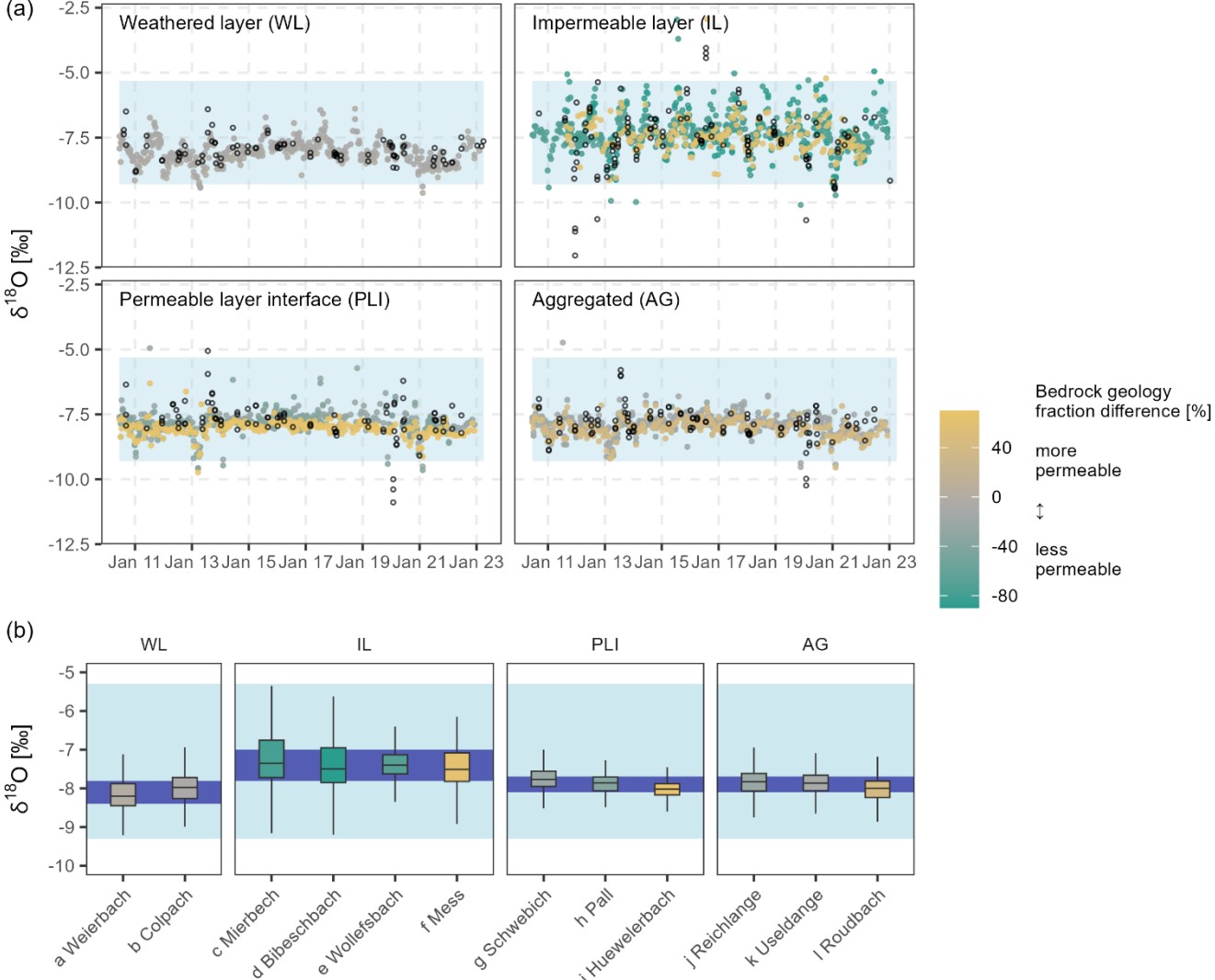

**Figure 5: (a)** Isotope signatures in streamflow from 2011 to 2023 for all 12 nested catchments grouped into hydro-lithological categories. Samples that were taken after strong rainfall events (i.e., more than 15 mm in the 72 hours preceding the sampling) are indicated by empty circles. **(b)** Boxplots of $\delta^{18}O$ in streamflow in all 12 nested catchments showing catchment-specific differences inside the hydro-lithological categories. The shadings indicate the interquartile range of precipitation (light blue) and streamflow (dark blue), displaying the damping of precipitation $\delta^{18}O$ in streamflow. The colour shading corresponds to the difference of the fraction of permeable bedrock (i.e., sandstone and conglomerates) minus the fraction of impermeable bedrock (i.e., marls and claystone).

 **3.3 Ensemble hydrograph separation**

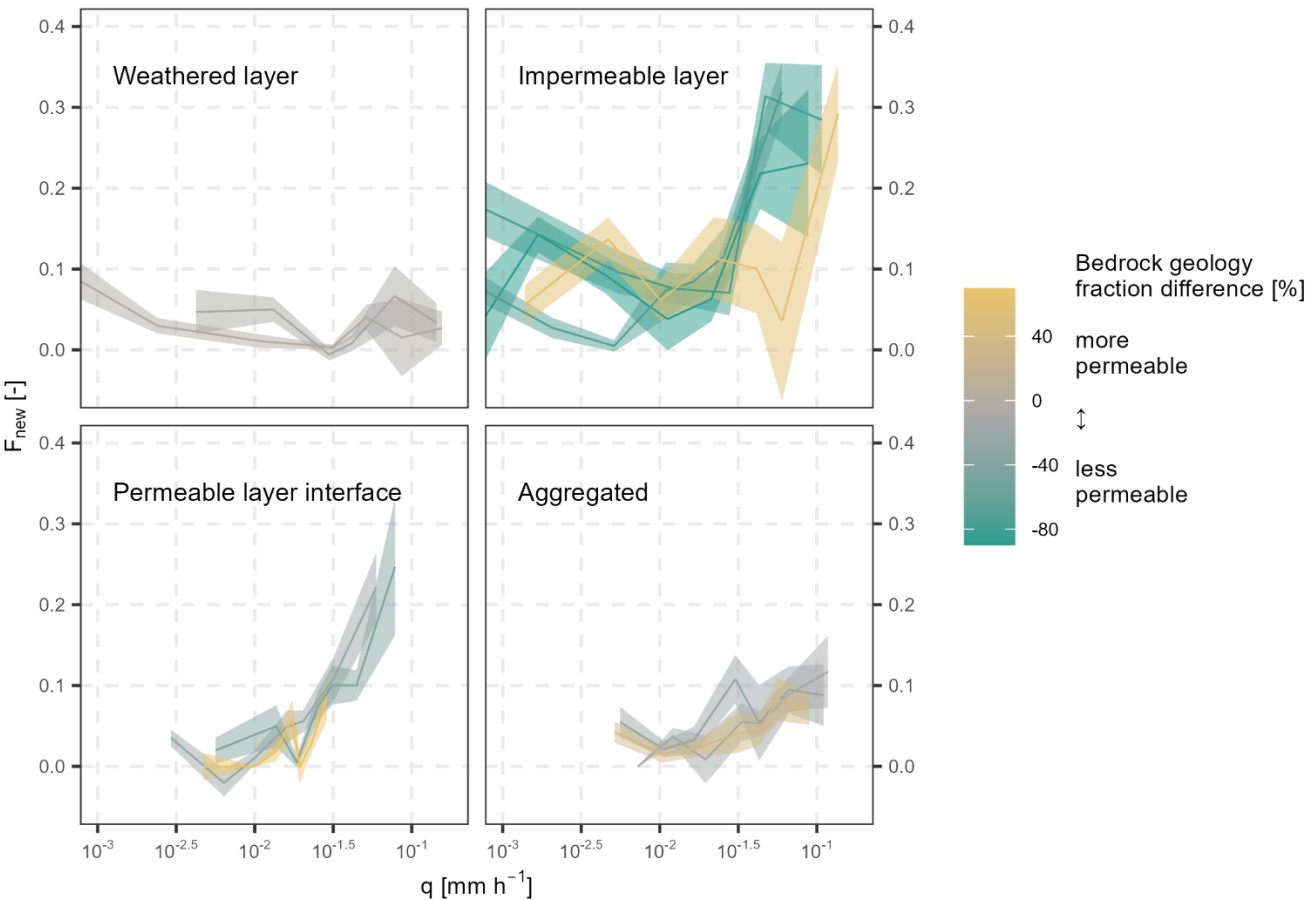

**Figure 6: Fraction of water younger than ~ 16 days ($F_{new}$) in the twelve nested catchments with increasing specific streamflow (taking the 24-hour means of specific streamflow before the timing of the grab samples) (lines, see Fig. B1 for the exact streamflow quantiles), including the standard errors (shading) regrouped into hydro-lithological categories. Mean $F_{new}$ was largest in the impermeable layer catchments and only substantially increased with higher mean daily specific streamflow in the impermeable layer and permeable layer interface catchments. The colour shading corresponds to the difference of the fraction of permeable bedrock (i.e., sandstone and conglomerates) minus the fraction of impermeable bedrock (i.e., marls and claystone).**

The estimated mean fraction of water younger than ~16 days ($F_{new}$) was largest in the impermeable layer catchments (Fig. 6; mean of 8.7%), ranging from 4.5% at Wollefsbach to 8.7% at Mess and 11.9% at Mierbech. Mean $F_{new}$ in the remaining catchments did not exceed 4.0%. In the permeable layer interface catchments mean $F_{new}$ was 2.1 % (2.2% at Schwebich, 2.7% at Pall and 1.3% at Huewelerbach); mean $F_{new}$ was 3.3% in the weathered layer catchments and 3.0% in the aggregated catchments.

$F_{new}$ increased with increasing streamflow in most catchments (and with 2-week antecedent precipitation, Fig. B2 in Appendix), except for the weathered layer catchments, where $F_{new}$ tended to be similar independent of streamflow magnitude (Fig. 6). The highest $F_{new}$ was at high streamflow in the Wollefsbach (up to ~ 35%). The impermeable layer catchments showed the greatest increase in $F_{new}$ with increasing streamflow (from a mean of ~ 9 to ~ 30%). In relative terms, increasing $F_{new}$ with increasing precipitation and streamflow was also evident in aggregated catchments (from a mean of ~ 3 to ~ 10%) and, most strikingly, in the permeable layer interface catchments (from a mean of ~ 2 to ~ 25%) (Figs. B1 and B2 in Appendix).

We further used the EHS method to estimate TTDs in the 12 nested catchments for longer time intervals. By taking the integral of the obtained TTDs, we were able to estimate $F_{new}$ for different, longer sampling intervals (i.e., time windows from 2 up to 26 weeks, Fig. 7). This analysis revealed that $F_{new}$ increased with the square root of the length of time interval in all catchments, reaching up to ~ 40% after 26 weeks. We also calculated $F_{new}$ for the upper 20% of streamflow and found that these high-flow $F_{new}$ were systematically higher compared to $F_{new}$ for all the streamflow data combined, except in the weathered layer

catchments and one aggregated catchment (Roudbach). The difference between $F_{new}$ calculated for overall conditions and high-streamflow conditions also tended to increase at longer intervals, e.g., for the marly sandstone Mess catchment, or the permeable layer interface catchments (but not the Huewelerbach). $F_{new}$ in the Wollefsbach plotted close to the mean of all catchments, contrasting with other impermeable layer catchments, which had substantially higher $F_{new}$ (Fig. 7e). $F_{new}$ for the upper 20% of streamflow increased with the interval length to a mean of 23.4 ± 3.0% at 26 weeks in the weathered layer

catchments, 46.6 ± 21.4% in the impermeable layer catchments, 18.9 ± 13.2% in the permeable layer catchments and 20.3 ± 9.0% in the aggregated catchments.

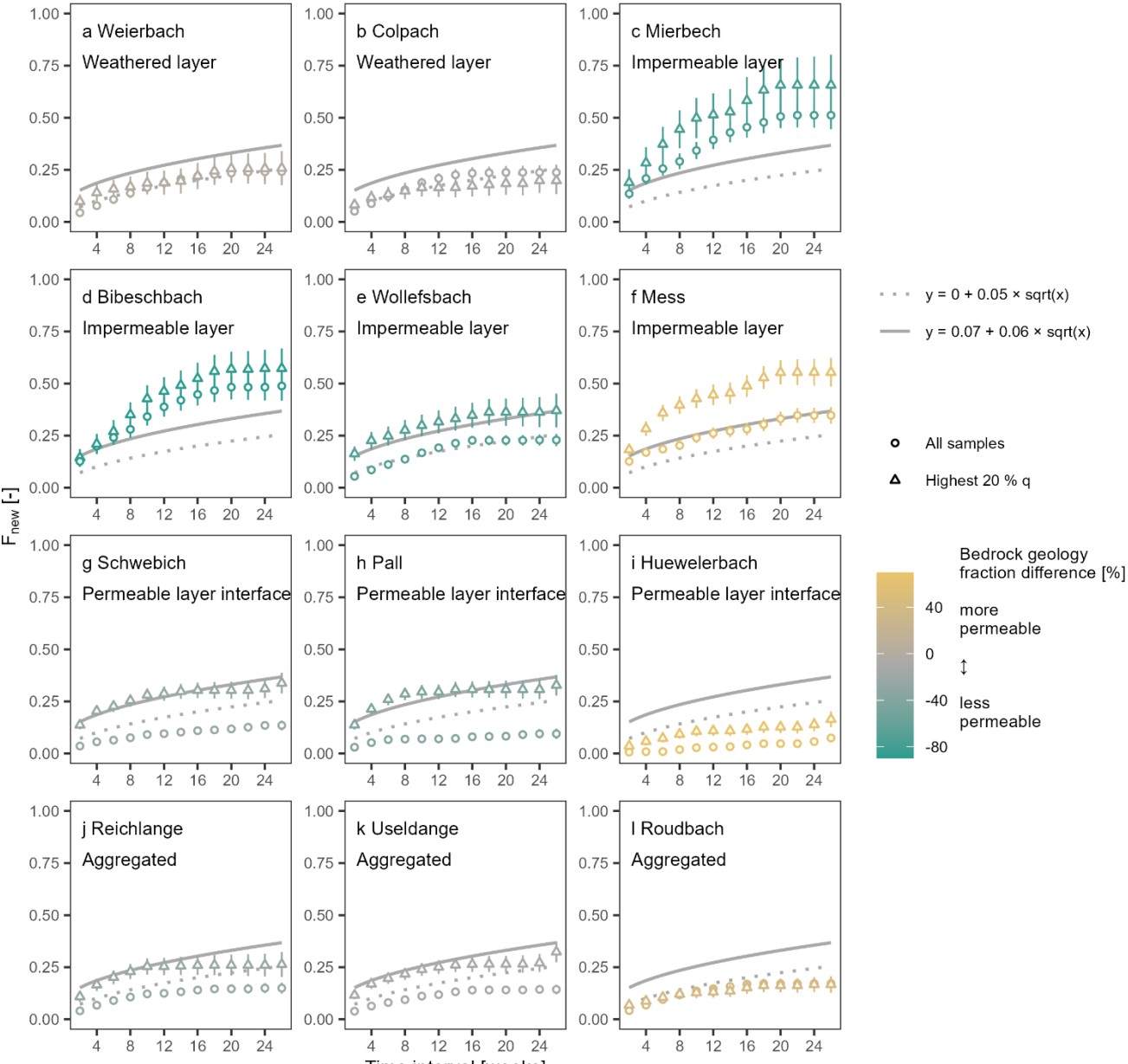

**Figure 7: Estimates of fractions of new water ($F_{new}$) and respective standard errors for sampling intervals of two to 26 weeks. The dotted lines indicate the mean for all 12 catchments based on the entire time series and the continuous lines indicate the mean for all**
440 **12 catchments for the highest 20% of specific streamflow; readers should note that these reference lines are the same in all panels. $F_{new}$ increased with increasing interval length and was systematically higher for the highest 20% of specific streamflow (triangles) than for the entire timeseries (circles) in the impermeable layer catchments, two of three aggregated catchments and permeable interface layer catchments. The colour shading corresponds to the difference of the fraction of permeable bedrock (i.e., sandstone and conglomerates) minus the fraction of impermeable bedrock (i.e., marls and claystone).**

## 3.4 Relationship between $F_{new}$ and catchment properties

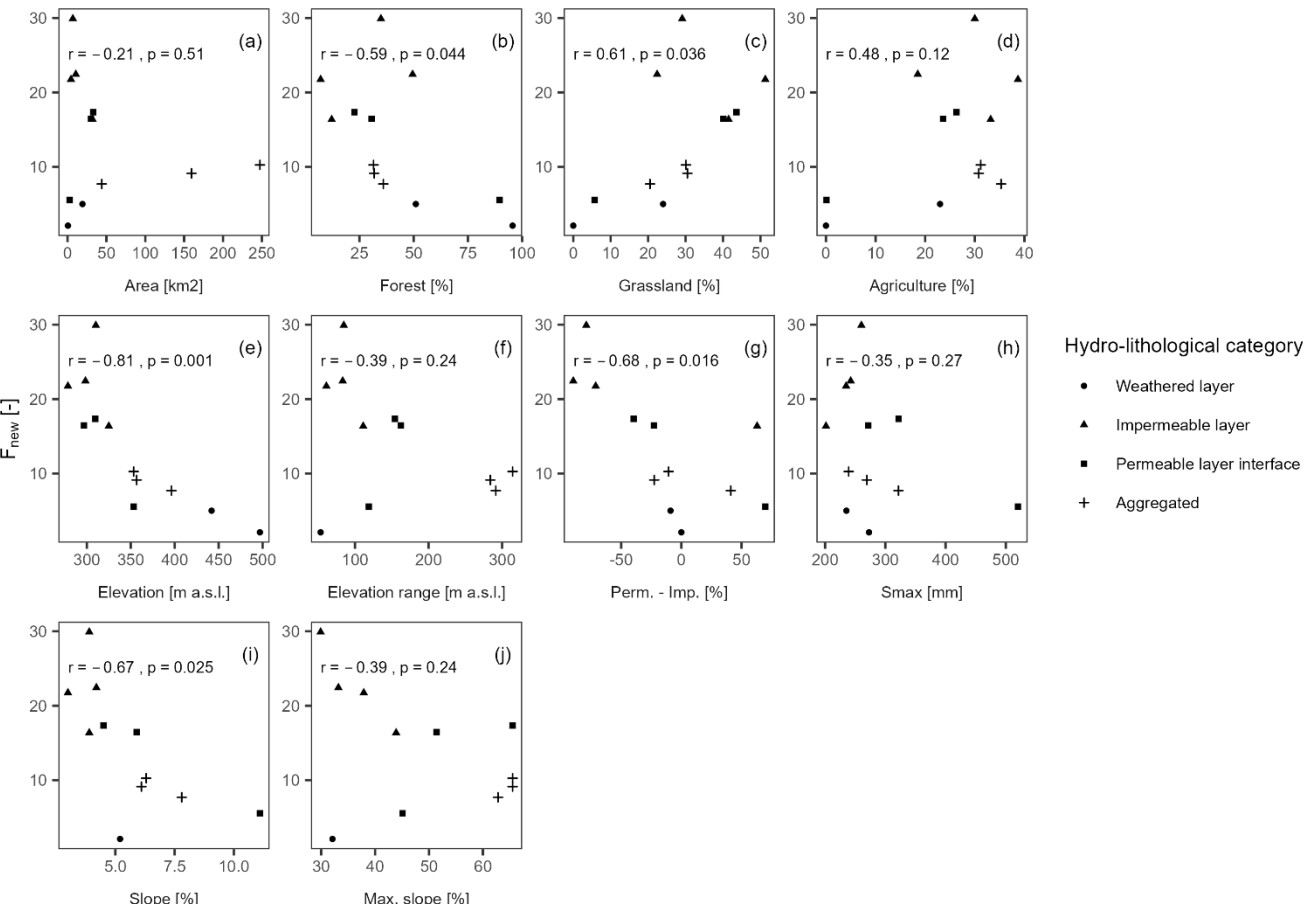

Figure 8: Scatter plots of the fractions of water younger than ~ 16 days ($F_{new}$) for the highest 20 % of specific streamflow by (a) catchment area, fractions of (b) forest, (c) grassland and (d) agriculture, (e) mean catchment elevation, (f) catchment elevation range, (g) impermeable to permeable bedrock fraction difference, (h) maximum storage capacity, (i) mean catchment slope, and (j) maximum catchment slope. Correlations (N = 12) are expressed in each plot with Pearson's r and corresponding $p$ (rounded after the third decimal). The elevation range (f), slope (i), and maximum slope (j) were missing for the Colpach catchment. At high streamflow rates (top 20 %), $F_{new}$ exhibited significant correlations with land use types, i.e., the fraction of forests (r = -0.59, p = 0.044) and grassland (r = 0.61, p = 0.036), mean catchment elevation (r = -0.81, p = 0.001), permeable to impermeable bedrock fraction difference (r = -0.68, p = 0.016), and mean slope (r = -0.67, p = 0.025). A correlation matrix of the physiographic catchment properties is shown in figure C1 (see Appendix).

Our results showed links between $F_{new}$ and physiographic catchment attributes. Along with bedrock geology, land use and catchment elevation had a substantial effect on the fraction of water younger than ~16 days. At high streamflow rates (top 20 %), $F_{new}$ exhibited significant correlations with land use types, i.e., the fraction of forests (r = -0.59, p = 0.036) and grassland (r = 0.61, p = 0.036), mean catchment elevation (r = -0.81, p = 0.001), permeable to impermeable bedrock fraction difference (r = -0.68, p = 0.016) ), and mean slope (r = -0.67, p = 0.025). There were no significant relationships between $F_{new}$ and catchment area (r = -0.21, p = 0.510), nor $S_{max}$ (r = -0.35, p = 0.270). However, we observed smaller $F_{new}$ variance with increasing catchment size (Fig. 8a). Choosing the elevation range instead of mean elevation did not yield more reliable results, likewise maximum slope instead of mean slope did not yield higher correlations (Fig. 8f and 8j), Note that bedrock geology, land use and catchment elevation were all significantly correlated to each other (Fig. C1 in Appendix).

## 4 Discussion

### 4.1 Bedrock geology effects on catchment response, new water fractions, and fast flow paths in nested catchments

Many years of hydrological studies in the Alzette River basin (e.g., Wrede et al., 2015; Martínez-Carreras et al., 2016; Pfister et al., 2017; Kaplan et al., 2022) have shown that bedrock geology has large effects on catchment hydrological behaviour, but the proportion of water travelling to the streams via flow paths with different transit times have not previously been quantified. Here we demonstrate a clear link between bedrock geology and catchment responses, isotope signatures and fractions of water younger than ~16 days ($F_{new}$) in 12 nested catchments of the Alzette River basin. In the impermeable layer catchments, median

streamflow $\delta^{18}O$ was nearly identical to $\delta^{18}O$ in precipitation (-7.4 ‰) suggesting that streamflow was well mixed between summer and winter precipitation, unlike the other catchment clusters where streamflow was dominated by winter precipitation. $F_{new}$ was highest in impermeable layer catchments (9.6 to 12.2%) and increased with higher streamflow in catchments with at least some sections of marly, impermeable bedrock ($F_{new}$ up to 45%, Figs. 6 and B1). This increasing $F_{new}$ behaviour was accordingly observed in permeable layer interface catchments (where sandstone and conglomerates are gradually replaced by

underlying marl and claystone layers), but absent in weathered layer catchments, despite their marked reactivity in streamflow response (Figs. 2 and 6). In weathered layer catchments, over 80% of the streamflow was older than 12 weeks (or three months), even at high streamflow (Fig. 7). In a similar study in Central Germany (23 sub-catchments), fractured bedrock (greywacke, schist, and granite) and freely draining soils in mountainous sub-catchments led to deep flow paths with long transit times, in contrast to rapid drainage with short transit times in the sedimentary bedrock lowlands (Lutz et al., 2018). The study yielded

fractions of young water (water less than 2 to 3 months old) from 1 to 27%, corroborating our results which lie in the same range (Fig. 7). The relationship between new water fractions and catchment properties has also been analysed by Floriancic et al. (2024a) across 32 Alpine catchments located in Austria and Switzerland, providing further evidence for physiographic controls on catchment behaviour. They found a significant negative correlation with catchment area, baseflow index, and terrain roughness.

Factors such as the bedrock depth, regolith thickness, or the presence of fractures or faults are important controls on subsurface storage and connectivity but were not explicitly considered here. The main reason for this was that such data were only available for specific catchments, where physiographic controls on catchment functions had been previously investigated in more detail, e.g., in the Weierbach, Wollefsbach, and Huewelerbach catchments (Wrede et al., 2015; Martínez-Carreras et al., 2016; Douinot et al., 2022; Kaplan et al., 2022). In these previous studies, the controls of subsurface structure on catchment

functioning were described, e.g., cracks and fissures in the schistose Weierbach catchment. Such subsurface structures encourage preferential flow paths and lead to substantial storage volumes, contributing to the dominance of subsurface flow processes (Angermann et al., 2017). We argue that the maximum storage capacity estimated in this study implicitly contains information on the bedrock depth and regolith thickness but the subsurface structure per se remains poorly characterized. For example, we did not measure the weathering degree of the regolith in the schistose Colpach catchment, nor the (re-)opening

of fissures in the loamy soils through drying and wetting cycles in the marly and claystone catchments in this study. These are highly variable factors, both in their spatial and temporal (in the case of drought-induced cracks) distribution, which are difficult to measure or to estimate. Yet these areas of increased infiltration, retention, or connectivity might represent very important controls on runoff generation and thus remain a limitation in this study. A thorough analysis combining isotope-based evidence with extensive field measurements, e.g., as done in a headwater catchment in the Swiss Alps (Leuteritz, et al.,

2025), would be a considerable step forward. In the same optic, future research could broaden the applicability of this study by identifying the physical parameters inherent to geology-controlled mechanisms (e.g., bedrock depth and porosity, regolith thickness, hydraulic conductivity gradients, etc.) and by integrating them into distributed hydrological models with a stronger physical basis. Such a process-based approach would enhance the simulation and prediction capabilities of models in ungauged basins or under changing environmental conditions.

Other important controls that have been reported to affect catchment functions include topography, precipitation, soil properties or vegetation (Von Freyberg et al., 2018; Lutz et al., 2018, Floriancic et al., 2024a). Pfister et al. (2002) investigated to what extent drainage density, catchment shape, catchment area, specific slope, percentage of less permeable substrate, and land use may control stormflow across 18 nested catchments in the Alzette basin. They concluded that the percentage of impermeable substrate was the main control factor. Also, in our study, the transport metrics derived from isotope data were correlated to bedrock geology (Fig. C1) and were similar across catchments with similar bedrock geology. At high streamflow rates (top 20 %), we found that isotopically inferred $F_{new}$ was significantly correlated to land use types, e.g., the fraction of forests (r = -0.59, p = 0.044), mean catchment elevation (r = -0.81, p = 0.001), and permeable to impermeable bedrock fraction difference (r = -0.68, p = 0.016; Fig. 8). Since the difference of mean catchment elevations was rather small (~200 m; Fig. 8e) and the elevation ranges similar (~100 m; Fig. 8f) in the smaller catchments, the correlation of $F_{new}$ with catchment elevation might result from the spatial distribution of the nested catchments or the correlations between elevation, geology, and land cover (Fig. C1). The negative correlation with forested area may relate to the geological properties, since the two catchments with the highest percentage of forested area also consist almost exclusively of schist or sandstone, which are generally associated with low $F_{new}$. Nonetheless one could also expect higher infiltration rates with root structures in forests, e.g., as reported in weathered layer catchments (Angermann et al., 2017), which could improve the interpretability of our results. Also, soil consolidation or artificial drainage in agricultural areas triggering fast overland and shallow subsurface flow (Loritz et al., 2017), could lead to higher correlations between $F_{new}$ and land use. We found a significant negative relation of $F_{new}$ at high streamflow rates only with the mean slope of the catchments (r = -0.67, p = 0.025). This might be the result of steeper catchments draining "older" water, as larger storage volumes can be found with larger elevation differences (Jasechko et al., 2016; von Freyberg et al., 2018; Floriancic et al., 2024b). But given the relatively mild elevation differences among our study catchments, another explanation could be that the impermeable layer catchments have less steep slopes than aggregated catchments, or the small sandstone-dominated Huewelerbach catchment. However, the discrepancy between results obtained with the mean and maximum slope values illustrates that these results are sensitive to the specific topographic metrics that are used. Finding the right metric can be challenging in those cases.

Like previous investigations of physiographic controls on catchment functions, this study also faced uncertainties related to the spatial heterogeneity in catchment attributes. Geological features of the Alzette River basin are spatially heterogeneous, impeding straightforward attributions to simplified geological units. This was notably the case in the marly sandstones of the Mess, a catchment assigned to the impermeable layer category (Fig. 5). Flow paths under 2 weeks were only activated at the highest streamflow, indicating fast infiltration and retention except during high-intensity rainfall (Fig. B2), which contrasts with the rest of the impermeable layer catchments. Another limitation was that catchments assigned to the same hydro-lithological category, e.g., permeable layer interface catchments or aggregated catchments, had widely varying percentages of marls and claystone (and, conversely, sandstone and conglomerates). Despite the processes being of a similar nature in these catchments, the "marly" characteristics of flashy streamflow responses and fast flow paths might especially affect catchments where marls are more represented. This could explain the relatively small effect of increasing streamflow on $F_{new}$ in the Roudbach, which contains a significantly lower fraction of marls and claystone (and higher fraction of sandstone and conglomerates) compared to the other aggregated catchments (Fig. B1). Other sources of uncertainty are related to measurements such as rain gauges that are not necessarily always representative of the rainfall over an entire catchment. Also, in the densely populated Alzette River basin anthropogenic infrastructure (e.g., drainage channels in the Wollefsbach or water captured from springs in the Huewelerbach) interferes with natural flow processes.

When discussing transit times, flow paths and $F_{new}$, it is important to consider the time scales and the definitions of words such as 'new' or 'fast'. $F_{new}$ inherently increases with longer time intervals used for the calculation, because those determine what counts as 'new'. In this study, $F_{new}$ is the fraction of water less than ~16 days old rather than 'event water' from individual precipitation occurrences. Thus, the fortnightly sampling limits the $F_{new}$ interpretation in terms of fast-response components,

i.e., faster than the defined ~16 days interval, which will be particularly important in highly reactive streams, e.g., in impermeable layer catchments with high contents of marls or claystone. We could observe that large precipitation events led to occasional rapid responses of the hydrograph (Fig. A2) and isotopic response, as quantified by $F_{new}$, increased with 2-week antecedent precipitation (Fig. B2). During high streamflow events, much of the isotopic response in these catchments, about 30 percent according to our study (Fig. 6), will occur within that two-week timeframe, which would be interesting to investigate at higher temporal resolution. Overland flow processes can occur very quickly, within hours or even minutes (Blöschl, 2022). This can occur when intense precipitation falls on soils with low infiltration capacities, either because they are saturated or very dry (Loritz et al., 2017). By documenting the high proportion of relatively fast flow components (occurring in less than ~16 days) in these marly or claystone catchments, our study identifies streams that would benefit from high-frequency sampling or monitoring campaigns. Conversely, our results also show that the fast flow components (occurring in less than ~16 days) in weathered layer catchments dominated by schistose or quartzite bedrock geologies have only minor relevance for the streamflow response ($F_{new}$ of ~3.5%). For any given time scale of 'new' water defined by the user, $F_{new}$ also reflects the type and geology of the catchment and streamflow (Fig. 7). Rather than analysing the streamflow response to individual precipitation events, our study compares catchment functions under varying conditions and in different physiographic settings.

## 4.2 Conceptualisation of catchment functions across varying streamflow for representative bedrock geologies

Based on our findings, we argue that the clear influence of bedrock geology on catchment travel times and storage-release dynamics is linked to the permeability of the underlying bedrock. Sarah et al. (2024) provide complementary evidence, demonstrating that the saturated hydraulic conductivity is an important control on baseflow contributions in high-altitude catchments. This corroborates a mechanistic link between bedrock permeability and subsurface water release patterns, reflecting our results with the bedrock geology as the primary control governing catchment storage-release dynamics. To illustrate this point, permeable layer interface catchments exhibit deep infiltration of event water, retention of event water in subsurface storage, and slow sub-surface lateral flows at the aquifer-aquiclude interface that eventually reach the stream with a substantial delay, although this delay is shorter for precipitation falling closer to the stream (Pfister et al., 2018). In the permeable layer interface catchments, we found increasing $F_{new}$ with higher streamflow – suggesting the activation of preferential flow paths, and probably also overland or shallow sub-surface flow when the infiltration capacity is exceeded in the lower near-stream section dominated by marly lithologies. In impermeable layer catchments, overland and shallow sub-surface flow dominate, even for small rain events (Douinot et al., 2022). This fast flow path component becomes even more important with the filling of the limited storage volume in these catchments, translating into high $F_{new}$ reaching the stream. The activation and de-activation of preferential flow paths depending on the volumes of water in storage after the wetting phase in winter, i.e., the "inverse storage effect", has already been reported from numerous other investigations using similar metrics (Von Freyberg et al., 2018; Harman, 2015; Rodriguez et al., 2018). Based on our findings, we propose a conceptual summary for how variability in $F_{new}$, i.e., fast flow paths contributions, in streamflow is predominantly controlled by bedrock permeability (Fig. 9).

Our finding that a substantial fraction of stream water in weathered layer catchments is many months old, at both high and low flows, highlights the dominance of storage and subsurface flow in the regolith layer. This occurs as a combination of high infiltration rates together with fast vertical velocities (Glaser et al., 2019; Scaini et al., 2017), resulting from high degrees of weathering in the regolith and deep root structures (Angermann et al., 2017). Fast lateral displacement (Glaser et al., 2019) in the weathered layer eventually leads to a significant fraction of streamflow being less than 12 weeks old (about 20%, see Fig. 7), despite the rest being older water and only a small fraction being less than 2 weeks old (about 3.3%). The retention of event water could additionally be caused by a "fill-and-spill" mechanism (Tromp-Van Meerveld and McDonnell, 2006) in fissures

filled by weathered material, explaining the delayed streamflow peak behaviour typical of weathered layer catchments. The schistose catchments in this study are indeed known to frequently shift between single and double peak hydrographs, mostly in response to alternating contributions from various landscape units and the exceedance of storage thresholds (Martínez-Carreras et al., 2016). The first peak was found to be the immediate response to rainfall dominated by flow paths close to the surface. The second, larger peak, taking days to build up, was controlled by delayed precipitation release from storage, e.g., as

demonstrated by a sprinkling experiment in the Weierbach (Scaini et al., 2018). The dominant exfiltration of groundwater (Glaser et al., 2016, 2020) even during times of high subsurface storage could possibly explain why mean and high-flow $F_{new}$ are similar (Fig. 7). Such dominant contributions of groundwater might be the reason why we could not find a systematic relationship between $F_{new}$ and streamflow magnitudes. High infiltration rates and substantial retention volumes in the weathered layers of bedrock have also been reported for explaining the somewhat counter-intuitive observation of small $F_{new}$

in higher-elevation catchments with 'impermeable' schistose bedrock formations (Lutz et al., 2018).

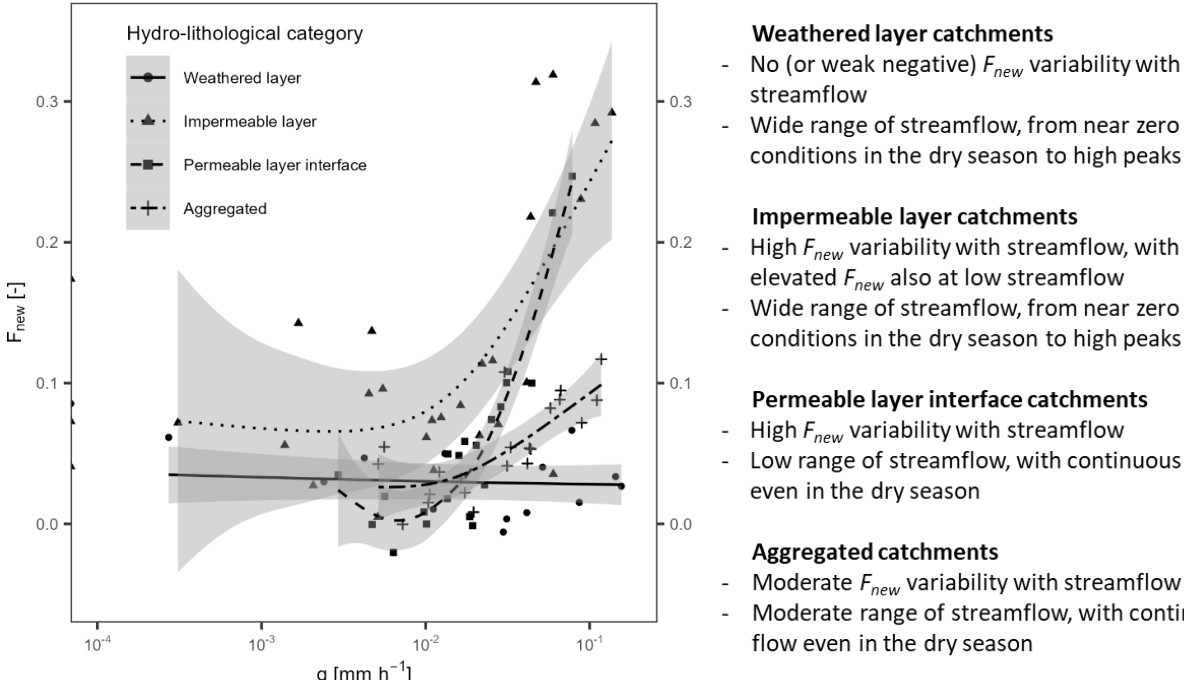

**Figure 9: Conceptual representation of water younger than ~ 16 days ($F_{new}$) with increasing specific streamflow in catchments from different hydro-lithological categories. Curves were fitted to $F_{new}$ disregarding the individual catchments, using a generalized**
**additive model (GAM) fitting algorithm with the *geom_smooth* function in R.**

### 4.3 Scale effects on time scales of catchment storage-release functions

River basins are intrinsically complex systems, exhibiting considerable spatial and temporal heterogeneity. With the spatial organisation of hydrologic dynamics being constantly (re)shaped by work performed by wetness and vegetation, we often fail

to capture the direct controls on the spatial and temporal heterogeneity of catchment functioning and evolution. In this context, rigid and static conceptualisations of catchments may be serious impediments to, e.g., climate mitigation and adaptation measures (Ehret et al., 2014). Despite considerable progress made in transit time estimation with new concepts proposed around time-variant transit and residence times (e.g., Botter et al., 2011; Benettin et al., 2022), upscaling from headwater catchments to mesoscale catchments to large drainage basins remains challenging. Soulsby et al. (2009) and Hrachowitz et al.

(2010) found that variation in MTT among 32 Scottish catchments (0.5–1700 km[2]) was highest for catchments smaller than 10 km[2]. In the Alzette River basin, Pfister et al. (2017) found the highest spread of MTTs for catchments smaller than 20 km[2],

with a mean MTT of ~ 1.4 years in the largest catchments, similar to the average of ~ 2 years found in the 32 Scottish catchments.

In the current nested catchment set-up (0.45 to 247.5 km$^2$), with contrasting physiographic settings and rather homogeneous climate, we found that bedrock geology controls fast flow paths. Major geological properties overlap with distinct hydrological functioning. In larger and less homogeneous catchments, these patterns tend to converge to a central behaviour, but small areas with discrete bedrock geologies can distinctively affect rainfall-runoff response. For example, fast flow paths activated in comparatively small marly headwaters may still shape the hydrograph further downstream by exceeding contributions from less responsive areas. Likewise, contributions from small permeable layer interface headwaters or sandstone sections may help sustain streamflow during dry spells in larger aggregated catchments (including less permeable and contributory sections). However, an apparent variance collapse in $F_{new}$ beyond ~ 150 km$^2$ (Figs. 6 and 8) suggests that the variability in flow path responses is dampened with increasing catchment size. The variance collapse in $F_{new}$ may also be attributed to the combined expression of increasing basin sizes and aggregated expressions of distinct regional physiographic features and associated flow path configurations.

## 4.4 Implications of our findings for predictions in ungauged basins and predicting catchment responses under non-stationary boundary conditions

For projecting hydrological responses under future climate scenarios and assessing catchment functions under non-stationary conditions, we can rely on the activation of fast flow paths with increasing rainfall intensities and streamflow, which appear to be linked to bedrock geology (Figures 6, B1 and B2). Our results suggest that an intensification of the hydrological cycle (Allen and Ingram, 2002; Blöschl et al., 2017) may lead to the most apparent changes in impermeable layer catchments, and probably significantly affect permeable layer interface catchments with at least some marl and claystone-dominated sections. For our study area, large storage capacities in the sandstone-dominated Huewelerbach catchment and high infiltration capacities in weathered schistose catchments (Angermann et al., 2017) also led to a strong buffering of fast flow paths during highly intense precipitation events, corroborating previous observations in schistose geologies (Lutz et al., 2018). In general, our results validate the mechanistic conceptualisation of fractions of young water (2 to 3 months old) relating to streamflow previously identified in the Swiss Alps, where $F_{new}$ is linked to catchment wetness, infiltration capacity and storage volume (Von Freyberg et al., 2018). The variance collapse in $F_{new}$ (younger than ~ 16 days) for catchments larger than 150 km$^2$ further illustrates that the transferability and reproducibility of our findings is limited to smaller catchments. Beyond the mesoscale, the variability in flow path responses might become smaller, but it will certainly remain affected by processes occurring in small headwater catchments, as shown by our findings (e.g., $F_{new}$ increasing with higher streamflow in aggregated catchments). Relatively small proportions of distinct bedrock types shape the overall catchment response to precipitation suggesting that more attention should be given to the most distinguishable sections of a catchment, instead of considering only spatially averaged properties across entire catchments.

## 5 Conclusions

With a view to better understand how landscape features translate into catchment functions driven by non-stationary forcings, we assessed the influence of bedrock geology on isotopically inferred fast flow paths in a set of 12 nested catchments. We found that the fractions of new water ($F_{new}$; water younger than ~16 days in this study) were linked to bedrock geology. $F_{new}$ was highest (up to ~ 45%) and increased most with streamflow, in impermeable layer catchments (i.e., dominated by marls and claystone) due to dominant overland and shallow sub-surface streamflow contributions. While increasing $F_{new}$ with increasing streamflow was also observed in permeable layer interface catchments (i.e., with high a fraction of sandstone and conglomerates with underlying marl and claystone layers) and linked to the contribution of downstream impermeable layer

sections, it was absent from weathered layer catchments (i.e., catchments dominated by schists and quartzites). In weathered layer catchments, a major fraction of water (~ 80%) is older than 12 weeks, with only a small fraction being less than 2 weeks old ($F_{new}$ of ~ 3.5%). For catchments smaller than 150 km$^2$, our results support the hypothesis that bedrock geology, linked to different storage-release dynamics, translates into quantifiable variations in fractions of fast flow paths inside the catchments. Catchments with similar bedrock geologies showed similar fast flow path patterns evolving with increasing rainfall intensities and streamflow.

The contrasting fast flow path behaviour of permeable and impermeable bedrock formations is an important finding for predictions of catchment functions under non-stationary conditions or in ungauged basins. According to our results, an intensification of the hydrological cycle would lead to the most apparent changes in impermeable layer catchments with an increased activation of fast flow paths, while also affecting permeable layer interface catchments that have large proportions of marls. The collapse in the variance of $F_{new}$ for catchments larger than 150 km$^2$ suggests an averaging effect through the aggregation of contrasted responses from different contributing areas. Sandstone catchments with large storage capacities and weathered layer catchments with high degrees of weathering showed only small contributions of fast flow paths to streamflow. These findings corroborate other inter-catchment studies in different settings and climates, suggesting that catchment sensitivity to a changing climate can be inferred from landscape features. Yet, while this study reveals systemic controls of bedrock geology on catchment travel times and storage-release dynamics, opening new avenues for large-scale applications, the application frontier needs to be further explored and broadened. The ~16-day window of isotope observations used in this study are only partially informative on event-scale processes. Also, the factors behind geology-controlled mechanisms have not been clearly identified. More isotope-based research investigating the influence of physical parameters on geology-controlled mechanisms is needed at higher resolutions to integrate those mechanisms into distributed hydrological models with a stronger physical basis. Such a process-based approach shall enhance simulation and prediction capabilities of models in ungauged basins or under changing environmental conditions. Success will depend on our ability to generalize these results and produce appropriately simplified geological maps adapted for hydrological applications at the macroscale. The currently limited hydrometric datasets remain another important constraint.

**Appendix A**

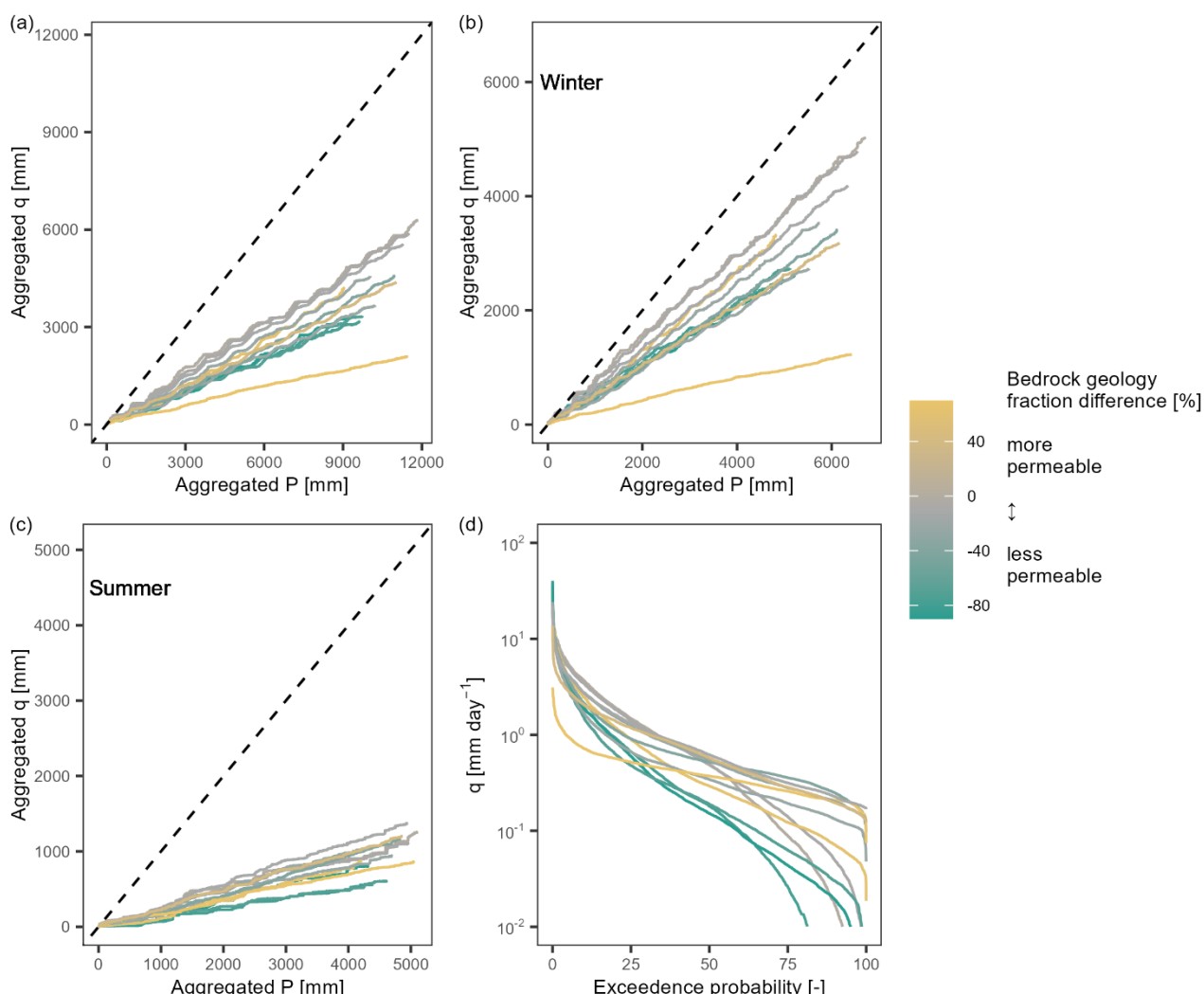

**Figure A1: Streamflow generation in the 12 nested catchments: (a) double mass curves of aggregated precipitation versus streamflow, (b) double mass curves for winter values only, (c) double mass curves for summer values only, and (d) flow duration curves of specific streamflow. The dotted lines in plots (a), (b) and (c) represent 1:1 lines. The colour shading corresponds to the difference of the fraction of permeable bedrock (i.e., sandstone and conglomerates) minus the fraction of impermeable bedrock (i.e., marls and claystone).**

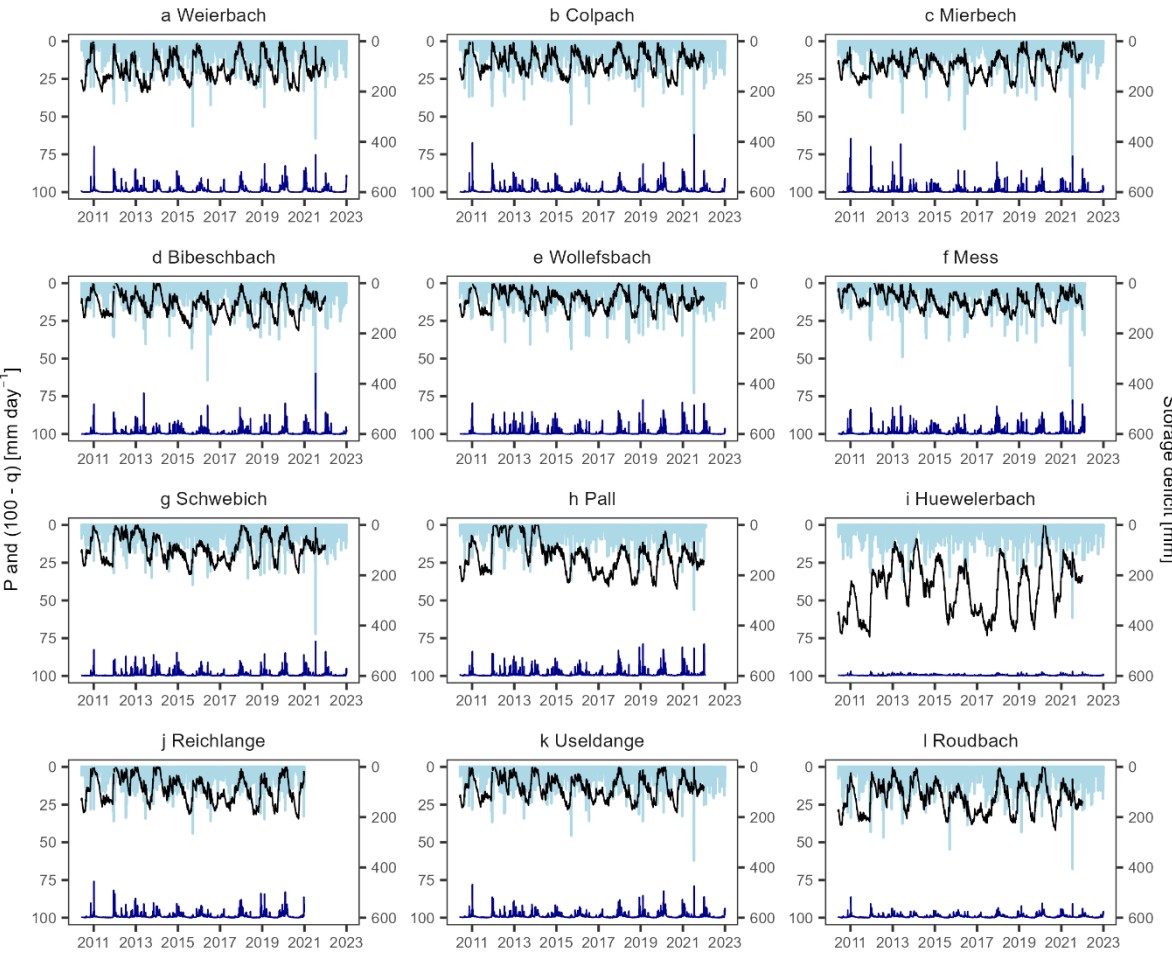

**Figure A2: Daily precipitation (P, light blue bars), specific streamflow (q, dark blue line), and storage deficit (D, black line), in the 12 nested catchments of the Alzette basin.**

**Appendix B**

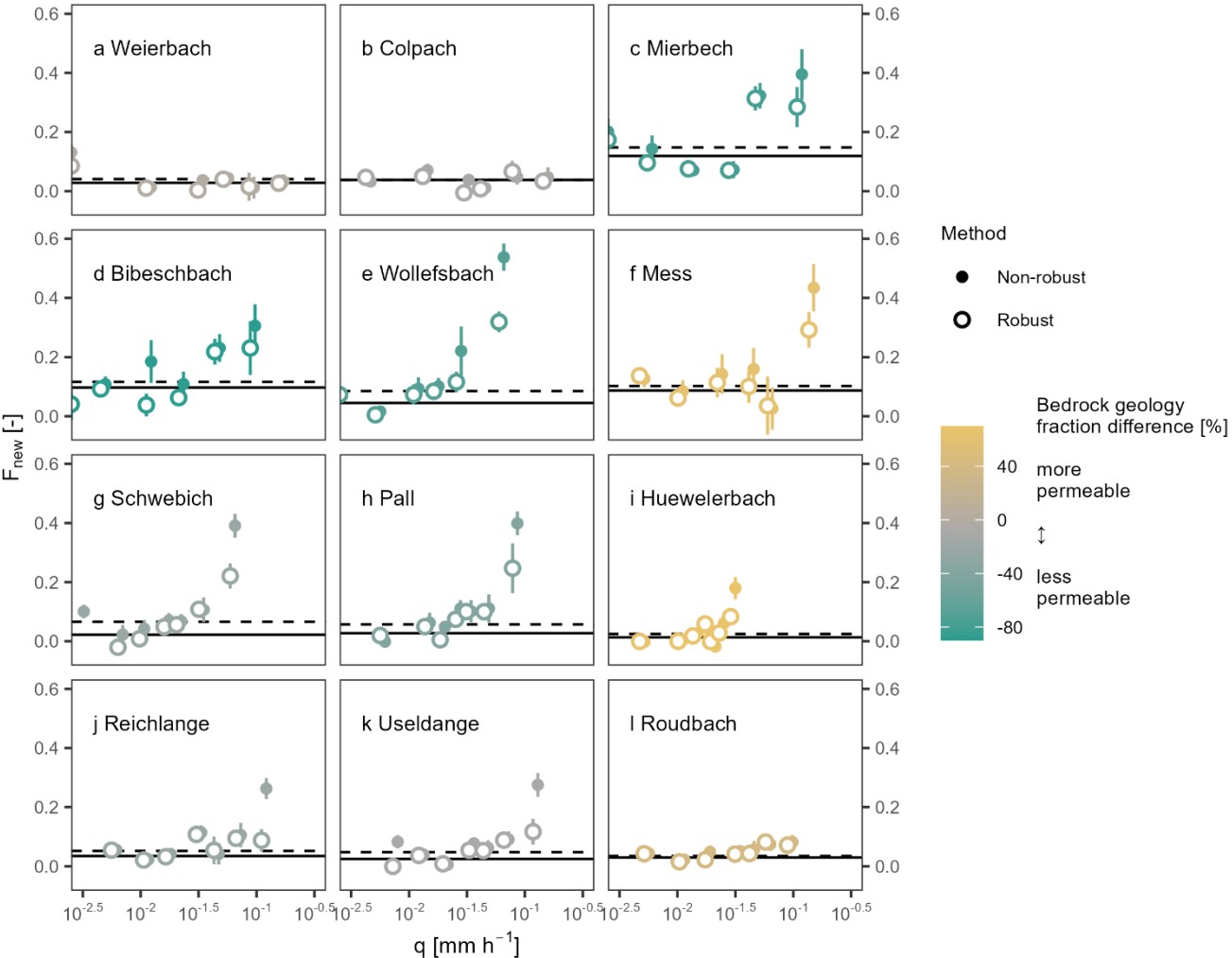

**Figure B1: Profiles of the fractions of water younger than ~ 16 days ($F_{new}$) for daily mean specific streamflow percentiles (0-20, 20-40, 40-60, 60-70, 80-90, 90-100), and respective standard errors, calculated using the robust (empty circles) and non-robust (full circles) solving algorithm from Kirchner (2019). The solid lines represent the average robust $F_{new}$ obtained for each of the catchments; the dotted lines were obtained with the non-robust method. The colour shading corresponds to the difference of the fraction of permeable bedrock (i.e., sandstone and conglomerates) minus the fraction of impermeable bedrock (i.e., marls and claystone).**

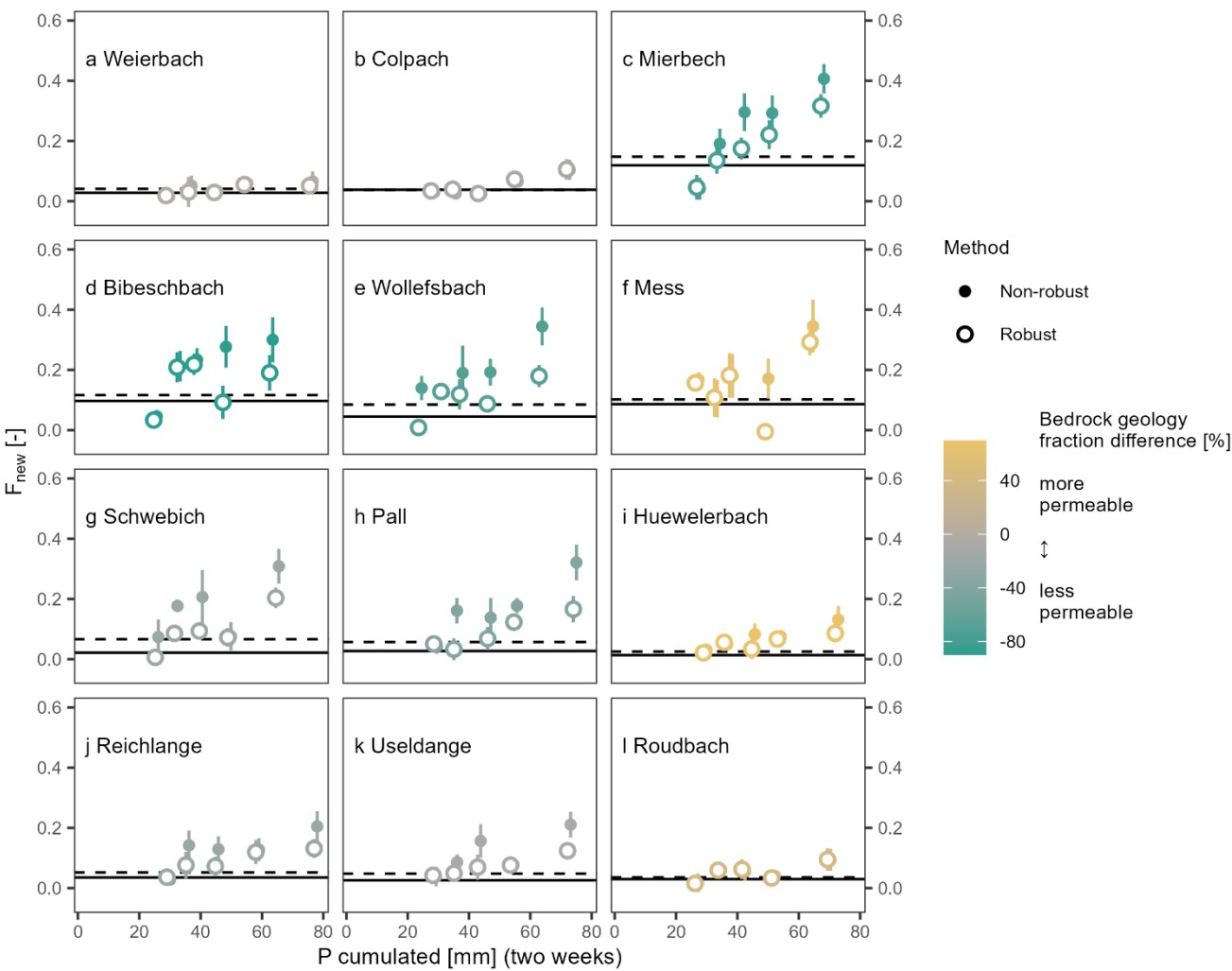

**Figure B2: Profiles of the fractions water younger than ~ 16 days ($F_{new}$) for precipitation deciles >50% during the two weeks, and respective standard errors, calculated with the robust (empty circles) and non-robust (full circles) estimation method. The solid lines represent the robust $F_{new}$ obtained for each of the twelve catchments; the dotted lines were obtained with the non-robust method. The colour shading corresponds to the difference of the fraction of permeable bedrock (i.e., sandstone and conglomerates) minus**

715 **the fraction of impermeable bedrock (i.e., marls and claystone). The results imply higher $F_{new}$ in catchments with impermeable bedrock geologies and an overall increase with increasing antecedent precipitation, which is more pronounced in catchments with a significant fraction of marls and claystone.**

**Appendix C**

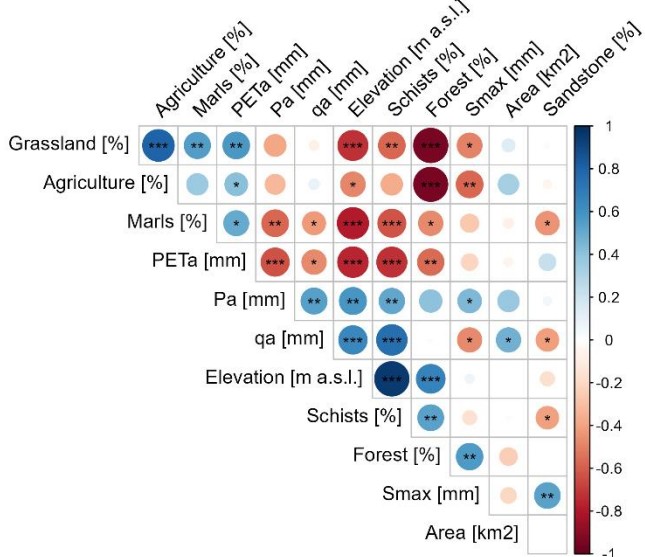

**Figure C1: Correlation matrix (Pearson's r, colour scale) of the parameters defining the physiographic characteristics of the nested catchments extracted from the CAMELS-LUX database (Nijzink et al., 2024) using the corrplot package in R (Wei and Simko, 2021). Significance levels (0.001, 0.01, 0.05) are indicated by the stars (***, **, *). Results show that land use, bedrock geology, catchment elevation, and hydrometric values in our set of nested catchments are statistically correlated to some degree. Only the fraction of sandstones, the catchment area and the maximum storage capacity appear to be less correlated to other catchment characteristics.**

## Data availability

The data that support the findings of this study are available from the corresponding author upon reasonable request.

## Author contribution

LP and BRS conceptualized the study in the framework of the MUSES project (Freshwater pearl mussels as stream water stable isotope recorders), funded by the German Research Fund, DFG, and the National Research Fund of Luxembourg, FNR (Grant C20/SR114757154/MUSES). LG collected the stream water and precipitation samples and curated the data; LL performed the isotope analyses. The study was conducted by GT, based on the methods developed by JWK. RK and MGF provided substantial support in designing the first draft of the manuscript, written by GT. The draft was thoroughly reviewed and edited by all co-authors.

## Competing interests

Some authors are members of the editorial board of journal Hydrology and Earth System Sciences.

**Acknowledgements**

We acknowledge support from J. F. Iffly, J. Juilleret, N. Martínez-Carreras, C. Tailliez and F. Barnich for their help during the sampling campaigns, sample analysis, and constructive comments.

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
