# Peer review of "Bedrock geology controls on new water fractions and catchment functioning in contrasted nested catchments"

_EGUsphere, 2025_

## Author Comment (AC1)

*Authors' response to Reviewer 1*
*[hess-2025-1530-RC1]*

*We thank the reviewer for his evaluation of our manuscript and his many helpful comments (hess-2025-1530). Below we address the reviewer's comments (full text) indented by arrows and coloured in blue. We appreciate the efforts by the reviewer, which will help to improve our manuscript.*

**General comments**

The authors define Fnew as water younger than approximately 16 days, based on their fortnightly sampling interval. While the authors note that fortnightly sampling limits Fnew resolution, a brief discussion on how the chosen sampling frequency may bias or underrepresent fast-response components would strengthen the interpretation of Fnew.

*→ You are right to mention that fortnightly sampling limits the Fnew interpretation in terms of fast-response components, which can represent an issue in particularly reactive streams, e.g., in impermeable layer catchments with high contents of marls or claystone. During high streamflow events, much of the hydraulic response will occur within that two-week window timeframe, which would be interesting to investigate at higher temporal resolution. Once either the infiltration capacity or the storage capacity of the soils in these catchments has been reached, overland flow processes can occur very quickly, within hours or even minutes. This could be essential for studies focusing on trigger mechanisms of intense short-term events, such as flash floods caused by highly intensive convective rains. This study could serve as a guide to identify streams that would benefit from high-frequency sampling or monitoring campaigns. We will make sure to include this aspect in the discussion.*

While the manuscript rightly identifies bedrock geology as a key control on flow partitioning and Fnew, the analysis is limited to areal lithological classification. Crucial structural factors—such as bedrock depth, regolith thickness, and the presence of fractures or faults—are not considered, despite their known influence on subsurface storage and connectivity. If such data are available, they should be integrated into the analysis; otherwise, their absence should be acknowledged as a limitation.

*→ Thank you for the pertinent remark. Factors such as the bedrock depth, regolith thickness, and the presence of fractures or faults, or even soil properties, are important controls on subsurface storage and connectivity, which were not explicitly considered here. The main reason for this was that such data was only available for specific catchments, where physiographic controls on catchment functions had been previously investigated in detail, e.g., in the Weierbach, Wollefsbach, and Huewelerbach catchments (Wrede et al., 2015; Martínez-Carreras et al., 2016; Douinot et al., 2022; Kaplan et al., 2022). The subsurface structure was mentioned as a control on catchment functions in these anterior studies, e.g., cracks and fissures in the schistose Weierbach catchment, creating substantial storage volumes and explaining the dominance of subsurface flow (Angermann et al., 2017). The results and conclusions from these studies are taken as reference and briefly summarized under section 2.1. entitled "Physiographic characterisation of the nested catchments".*
*Another argument is that we estimated the maximum storage capacity of the catchments, which implicitly contains information on the bedrock depth and retention volume, or regoltih thickness. One could also argue that these factors will matter less in catchments where the infiltration rates are low and the bedrock is impermeable, i.e., marly and claystone catchments. They are however important controls in the schistose catchment with a highly weathered front, which might have a thickness that*

*varies due to the presence of fractures and fissures. We will acknowledge this as a limitation, particularly in the Colpach catchment, where the depth of the regolith is largely unknown.*

The manuscript emphasizes the dominant role of bedrock geology in shaping hydrological response, however, the potential confounding influence of co-varying catchment attributes (e.g., slope, land cover, elevation) is not adequately resolved.

*→ Thank you for bringing this to our attention. The confounding influence of co-varying catchment attributes was addressed under section 4.1. in the following paragraph:*

*"Other important controls that have been reported to affect catchment functions include topography, precipitation, soil properties or vegetation (Von Freyberg et al., 2018; Lutz et al., 2018, Floriancic et al., 2023), yet in our study area, many of these parameters were correlated (Fig. S5) or similar across the different catchments. Across the 12 examined catchments the underlying bedrock geology played an important role in their shaping landscapes thereby affecting topography, soil properties or vegetation. , we found that isotopically inferred $F_{new}$ during times of high streamflow was positively correlated to the percentage of marls (r = 0.88, p = 0.001), and to the percentage of grassland area (r = 0.61, p = 0.036, Fig. 8). The negative correlation with forest area (r = - 0.59, p = 0.044) may also relate to the geological properties, since the two catchments with the highest percentage of forest area also consist almost exclusively of schist or sandstone, which are generally associated with low Fnew. One could expect higher infiltration rates with root structures in forest, e.g. as reported in weathered layer catchments (Angermann et al., 2017), and soil consolidation or artificial drainage in agricultural areas triggering fast overland and shallow subsurface flow (Loritz et al., 2017), leading to correlations between $F_{new}$ and forest and agriculture land use."*

*We could also refer to Pfister et al. (2002), where they had already investigated to what extent factors such as drainage density, catchment shape, catchment area, specific slope, percentage of less permeable substrate, and land use may control stormflow across 18 nested catchments in the Alzette basin. It turned out that the percentage of impermeable substrate was the main control factor.*

*Although we acknowledge that forests could have an influence on the streamflow here, we could stress the importance of land use more. Upon second examination, a clear linear relation exists between grassland, or forest fractions, and the fractions of new water at high streamflow rates – if we disregard the two points with the highest $F_{new}$ (Fig. 8). We will mention that. The two points correspond to impermeable layer catchments, where one could argue that $F_{new}$ at high stream flow rates can be expected to be high, no matter the land use. Also, we will remove the statement "Considering only characteristics that were independent from each other" because it was motivated based on the results of the correlation matrix in Figure S5, which require a cautious interpretation as there were only 12 points (i.e., the number of nested catchments). Our argument was that the underlying bedrock geology contributes to shape catchment attributes, but it remains interesting to show and discuss the influence of these attributes following your advice. Thus, we propose to expand Figure 8 and add the elevation range, the mean slope and the maximum slope of the catchments, to include the topography in our discussion. Results suggest while a relation with the mean catchment elevation seems to exist (Fig. 8e), there is none with the elevation range. This seams to be an indication that the previous relation with mean elevation is just the result of the spatial distribution of the catchments with similar properties, as the hydro-lithological clusters appear clustered (Fig. 8e). For the slope, we find a significant negative relation of the fraction of new water at high streamflow rates with the mean slope of the catchments only. This result is rater surprising, as one would expect higher $F_{new}$ with steeper slopes. One explanation is that the impermeable layer catchments in our study have slopes that are less steep than aggregated catchments, or the small sandstone-dominated Huewelerbach catchment. However, the discrepancy between mean and maximum slope values already shows that these results are strongly dependent on how these parameters are computed.*

The manuscript would benefit from a clearer explanation of the methodological constraints of the Ensemble Hydrograph Separation (EHS) approach. Specifically, does EHS impose minimum requirements on sampling interval, time series length, or end-member stability?

*→ By definition, EHS assumes that the sampling interval is constant, as it compares isotopic concentrations at a certain timestep with previous observations without considering the exact moment of sampling. Still, this constraint is not very strict in the sense that EHS can still be applied to timeseries with slightly unequal timesteps without becoming unstable. In this case, the best approximation is to take the mean sampling interval as the definition for the fraction of new water, as was done in this study. Gaps in isotopic measurements need to be filled with NA values, while zero values can be filled in for missing streamflow or precipitation values, without affecting the computations. This step is important, because it affects the calculations if EHS assumes a direct link between two consecutive samples in the record, which are few sampling intervals apart. There is no minimum requirement for the timeseries length per se, however, larger datasets yield more significant results, with the error margins becoming smaller in comparison to the magnitude of the detected signal in either the $F_{new}$ profiles or the TTDs. The size of the dataset also becomes increasingly important when sub-setting the data for the computation of profiles or TTDs at different states of the catchment. For the profiles, we tried to work with at least ~30 observations, but in the end, that decision is in the hands of the user. The greatest issues we faced were when the order of the observations was not respected, e.g., by making mistakes in alignments of the streamflow and precipitation isotopic samples. Otherwise, the EHS method appears to be quite robust and broadly applicable. Please refer to Kirchner et al. (2019) to read the underlying principles of EHS, or Kirchner and Knapp (2020) for the practical guide on how to use EHS, with scripts provided in R and Matlab, and instructions on data requirements.*

**Specific comments**

Line 163: The phrase "i.e.i.e." is repeated. Please delete the redundant "i.e." to correct the typo.

*→ Thank you, we will remove it.*

Line 221: Please clarify how α limits evaporation.

*→ "Evapotranspiration" is probably the correct term here, as α will predominantly represent the reduction of transpiration with decreasing water availability. We hope this answers your question.*

Line 224: The manuscript defines Smax as the highest 0.5% of daily catchment storage values, please justify why this specific quantile was chosen

*→ We thought taking a quantile would be more robust than taking the absolute maximum of storage values, but we must admit that there was no specific reason other than that. We might consider taking the maximum value instead.*

---

## Author Comment (AC2)

*Authors' response to Reviewer 2*
*[hess-2025-1530-RC2]*

*We thank the reviewer for his evaluation of our manuscript and his many helpful comments (hess-2025-1530). Below we address the reviewer's comments (full text) indented by arrows and coloured in blue. We appreciate the efforts by the reviewer, which will help to improve our manuscript.*

**General comments**

Section 2.5 clearly outlines the EHS method, but its reliance on equations may pose difficulties for readers unfamiliar with isotope-based hydrograph separation. I recommend adding a simplified conceptual diagram or schematic to illustrate the EHS workflow.

→ *We acknowledge that it might be difficult to understand the EHS workflow from the equations we provide, especially for readers unfamiliar with isotope-based transit time estimations. However, one must consider that this paper will primarily target an audience well-familiar with these techniques – other readers will be more interested in the conclusions we draw based on EHS. This is why we think that this manuscript will not necessarily benefit from the addition of such an illustration, also because the EHS methodology is already well documented in Kirchner (2019) and Kirchner and Knapp (2020). Note that a scatter plot of $C_{Q,j}$ - $C_{Q,j-1}$ versus $C_{P,j}$ - $C_{Q,j-1}$ with the regression line representing the fraction of new water $F_{new}$ is presented in Kirchner and Knapp (2020), which is a very good introduction to the EHS framework.*

I recommend that the authors provide a brief rationale or literature-based justification for choosing the 16-day threshold as the representative time scale for defining "new water." It would also be helpful to clarify whether and to what extent this threshold might influence the interpretation of the results and the robustness of the study's conclusions.

→ *Thank you for the suggestion, the answer to your comment overlaps one of our answers to reviewer 1: the fortnightly sampling limits the Fnew interpretation in terms of fast-response components, which can represent an issue in particularly reactive streams, e.g., in impermeable layer catchments with hight contents of marls or claystone. During high streamflow events, much of the hydraulic response will occur within that two-week window timeframe, which would be interesting to investigate at higher temporal resolution. Once either the infiltration capacity or the storage capacity of the soils in these catchments has been reached, overland flow processes can occur very quickly, within hours or even minutes. This could be essential for studies focusing on trigger mechanisms of intense short-term events, such as flash floods caused by highly intensive convective rains. This study could serve as a guide to identify streams that would benefit from high-frequency sampling or monitoring campaigns. We will make sure to include this aspect in the discussion.*
*As to why the 16-day threshold was chosen for defining "new water", the 13-year record of isotopic measurements must be put in its original context. When the sampling started, EHS had not yet been developed and instead studies would rely on metrics such as the mean transit time, derived from convolution or sine wave fitting approaches. We briefly mention these techniques in the introduction:*

   *"However isotope-based studies have often relied on convolution or sine wave fitting approaches that are not well suited to capture the spatial and temporal heterogeneities that dominate streamflow generation in most catchments (Kirchner, 2016a, b). A common source of bias is a priori conjectures concerning the shape of the TTD (Remondi et al., 2018), resulting in, e.g., increasing uncertainty in mean transit time (MTT) estimates when MTT exceeds several years (DeWalle et al., 1997). More recently, calculations of the fraction of young water (Kirchner, 2016b) and transit times*

*extracted from storage selection functions (SAS) (Benettin et al., 2015; Harman, 2015; Rinaldo et al., 2015) have been proposed as more robust methods than traditional MTT estimates."*
*Often, monthly data would suffice to calculate metrics such as the fractions of young water, i.e., water travelling to the steams in less than 2-3 months (Kirchner, 2016), e.g., as has been done in analogous inter-catchment studies in Germany (Lutz et al., 2018) and in Switzerland (Von Freyberg et al., 2018), relying on fortnightly to monthly isotope data. Hence, for the techniques of the time, fortnightly isotopic measurements in multiple catchments were already state-of-the-art, high-resolution datasets. In this context, moving from the previous definition of fractions of water less than 2-3 months to fractions of water less than ~16 days old is already a considerable step. Of course, it would be advantageous to move to higher frequencies in catchments identified here, which we will consider for future investigations.*

The assumption of a 200 mm threshold for field capacity warrants further clarification. I recommend that the authors briefly justify whether this value reflects region-specific soil and climatic conditions, and whether it is based on measured soil data or literature from comparable settings. Clarifying this point would enhance the robustness of the catchment storage estimates.

*→ Again, thank you for the suggestion. For the threshold of 200 mm for the field capacity, we can refer to Pfister et al. (2017), as they have already done these storage calculations for the same catchments in the past, but for shorter periods. They assessed the sensitivity of the storage estimates to different values of the field capacity (100, 200, 300 mm) and found the daily offsets to be largely unaffected by the value chosen for the field capacity. Consequently, although the absolute storage estimates might differ, the storage deficit, ultimately used in this study, remains unchanged.*

Please note that the abbreviation "i.e." appears redundantly in both line 163 and line 253. Please delete the redundant "i.e.".

*→ Thank you, we will remove it.*

---

## Referee Report (RR1)

I have reviewed the authors' response to my previous comments, and I appreciate the thoughtful revisions and clarifications provided. The manuscript has been strengthened in several aspects. There remain, however, two key suggestions for further improvement.

1. It is suggested to add a paragraph on the future research direction in the Discussion or Conclusion section to highlight the application potential and frontier of this study. Specifically, this study reveals the systematic control of bedrock geology on catchment hydrological functions (such as the "new water" fraction, storage-release dynamics) and establishes a clear geology-function classification framework. The authors could point out that future research could parameterize such geology-controlled mechanisms (e.g., differences in permeability of different lithologies, thickness of the weathered layer, etc.) and integrate them into distributed hydrological models with a stronger physical basis, thereby enhancing the simulation and prediction capabilities of models in ungauged basins or under changing environmental conditions.

2. To further enhance the academic depth and foster a closer dialogue with current research frontiers, it is recommended that the authors incorporate citations to the following highly relevant and recent studies in the appropriate sections of their manuscript.

In the first paragraph of the Introduction, the authors rightly point out that an incomplete understanding of the complex interactions within subsurface hydrological systems represents a key obstacle to accurate streamflow prediction. To situate this assertion within a concrete and contemporary research context, I recommend citing the recent work by Ren et al. (2024). Their study provides a compelling quantitative example: in a karst watershed, explicitly accounting for the non-closure of underground catchment boundaries—where the effective contributing area dynamically expands with antecedent rainfall—and the spatial heterogeneity of subsurface water storage capacity can enhance the accuracy of streamflow predictions.

Reference: Ren, Z. L., Li, B. Q., Xiao, Y., Li, K.: Investigating spatial heterogeneity of karst water storage capacity and nonclosure of underground watersheds in karst hydrological simulation. Hydrol. Process., 38(12), e70012, https://doi.org/10.1002/hyp.70012.

In the first paragraph of the Introduction, the manuscript correctly identifies the persistent challenge of hydrological prediction in ungauged or dynamically changing catchments. To further strengthen this point by demonstrating a cutting-edge response to this exact challenge, I suggest citing Ye et al. (2024). Their study on a regionalization-strategy-guided LSTM model for flood forecasting in ungauged catchments provides a direct, contemporary example of how the hydrological community is tackling this issue with advanced methods.

Reference: Ye, K. J., Liang, Z. M., Chen, H. Y., Qian, M. K., Hu, Y. M., Bi, C. L., Wang, J., Li, B. Q.: Regionalization strategy guided long short-term memory model for improving flood forecasting. Hydrol. Process., 38(10), e15296, https://doi.org/10.1002/hyp.15296.

I suggest citing the recent study by Floriancic et al. (2024) in the Discussion section. This paper systematically analyzes the relationship between new water fractions and catchment properties across 32 Alpine catchments and finds a significant negative correlation with catchment area, baseflow index, and terrain ruggedness. Citing this work would strongly support and reinforce the conclusions of your study.

Reference: Floriancic, M. G., Stockinger, M. P., Kirchner, J. W., Stumpp, C.: Monthly new water fractions and their relationships with climate and catchment properties across Alpine rivers. Hydrol. Earth Syst. Sci.,28, 3675-3694, https://doi.org/10.5194/hess-28-3675-2024.

I also recommend citing Sarah et al. (2024), which provides critical complementary evidence.

They demonstrate that saturated hydraulic conductivity (Ksat) is the "key control" on baseflow contribution in high-altitude catchments.  This finding directly corroborates and quantifies the mechanistic link between bedrock permeability (which Ksat fundamentally reflects) and subsurface water release patterns, strongly supporting the authors' central thesis that geology is the primary filter governing catchment storage-release dynamics.

Reference: Sarah, S., Shah, W. S., Somers, L. D., Deshpande, R. D., Ahmed, S.: Saturated hydraulic conductivity (Ksat) and topographic controls on baseflow contribution in high-altitude aquifers with complex geology. J. Hydrol., 641, 131763, https://doi.org/10.1016/j.jhydrol.2024.131763.

---

## Author Response (AR2)

*Authors' response to Reviewer 1*

*We thank the reviewer for his/her evaluation of our revised manuscript and accepting it as it currently is. We appreciate the efforts by the reviewer, which helped to improve our manuscript.*

*Authors' response to Reviewer 2*

*We thank the reviewer for his/her evaluation of our revised manuscript and his/her helpful comments. Below we address the reviewer's comments (full text) indented by arrows and coloured in blue. We appreciate the efforts by the reviewer, which will help to improve our manuscript.*

**General comments**

1. It is suggested to add a paragraph on the future research direction in the Discussion or Conclusion section to highlight the application potential and frontier of this study. Specifically, this study reveals the systematic control of bedrock geology on catchment hydrological functions (such as the "new water" fraction, storage-release dynamics) and establishes a clear geology-function classification framework. The authors could point out that future research could parameterize such geology-controlled mechanisms (e.g., differences in permeability of different lithologies, thickness of the weathered layer, etc.) and integrate them into distributed hydrological models with a stronger physical basis, thereby enhancing the simulation and prediction capabilities of models in ungauged basins or under changing environmental conditions.

→ *Thank you for the suggestion, highlighting the application potential and frontier of the study is an important improvement of the manuscript. We have integrated your suggestion in the second paragraph of the Discussion (lines 505-509 of the track-change manuscript) and created a new paragraph in the Conclusion (lines 690-698).*

2. To further enhance the academic depth and foster a closer dialogue with current research frontiers, it is recommended that the authors incorporate citations to the following highly relevant and recent studies in the appropriate sections of their manuscript.
In the first paragraph of the Introduction, the authors rightly point out that an incomplete understanding of the complex interactions within subsurface hydrological systems represents a key obstacle to accurate streamflow prediction. To situate this assertion within a concrete and contemporary research context, I recommend citing the recent work by Ren et al. (2024). Their study provides a compelling quantitative example: in a karst watershed, explicitly accounting for the non-closure of underground catchment boundaries—where the effective contributing area dynamically expands with antecedent rainfall—and the spatial heterogeneity of subsurface water storage capacity can enhance the accuracy of streamflow predictions.
Reference: Ren, Z. L., Li, B. Q., Xiao, Y., Li, K.: Investigating spatial heterogeneity of karst water storage capacity and nonclosure of underground watersheds in karst hydrological simulation. Hydrol. Process., 38(12), e70012, https://doi.org/10.1002/hyp.70012.

→ *Thank you for the lead, we cite the recommended article (lines 36-39) in the first paragraph of the Introduction of the revised manuscript.*

In the first paragraph of the Introduction, the manuscript correctly identifies the persistent challenge of hydrological prediction in ungauged or dynamically changing catchments. To further strengthen this point by demonstrating a cutting-edge response to this exact challenge, I suggest citing Ye et al. (2024). Their study on a regionalization-strategy-guided LSTM model for flood forecasting in ungauged catchments provides a direct, contemporary example of how the hydrological community is tackling this issue with advanced methods.

Reference: Ye, K. J., Liang, Z. M., Chen, H. Y., Qian, M. K., Hu, Y. M., Bi, C. L., Wang, J., Li, B. Q.: Regionalization strategy guided long short-term memory model for improving flood forecasting. Hydrol. Process., 38(10), e15296, https://doi.org/10.1002/hyp.15296.

*→ We cite the recommended article (lines 45-45) in the first paragraph of the Introduction of the revised manuscript.*

I suggest citing the recent study by Floriancic et al. (2024) in the Discussion section. This paper systematically analyzes the relationship between new water fractions and catchment properties across 32 Alpine catchments and finds a significant negative correlation with catchment area, baseflow index, and terrain ruggedness. Citing this work would strongly support and reinforce the conclusions of your study. Reference: Floriancic, M. G., Stockinger, M. P., Kirchner, J. W., Stumpp, C.: Monthly new water fractions and their relationships with climate and catchment properties across Alpine rivers. Hydrol. Earth Syst. Sci.,28, 3675-3694, https://doi.org/10.5194/hess-28-3675-2024.

*→ We were in fact referring to the preprint version of that same article in the previous version of the manuscript (line 100 of the old manuscript). We changed it to the actual version of the published article, thank you for pointing this out.*

I also recommend citing Sarah et al. (2024), which provides critical complementary evidence. They demonstrate that saturated hydraulic conductivity (Ksat) is the "key control" on baseflow contribution in high-altitude catchments. This finding directly corroborates and quantifies the mechanistic link between bedrock permeability (which Ksat fundamentally reflects) and subsurface water release patterns, strongly supporting the authors' central thesis that geology is the primary filter governing catchment storage-release dynamics. Reference: Sarah, S., Shah, W. S., Somers, L. D., Deshpande, R. D., Ahmed, S.: Saturated hydraulic conductivity (Ksat) and topographic controls on baseflow contribution in high-altitude aquifers with complex geology. J. Hydrol., 641, 131763, https://doi.org/10.1016/j.jhydrol.2024.131763.

*→ This is indeed a great citation to corroborate our thesis on bedrock geology controls on catchment functions. We cite the recommended article (lines 570-573) in the first paragraph of second chapter of the Discussion of the revised manuscript.*